

November 22, 2016

# The radiative role of ozone and water vapour in the temperature annual cycle in the tropical tropopause layer

Alison Ming[1], Amanda C. Maycock[2], Peter Hitchcock[3], and Peter Haynes[1]

[1]Department of Applied Mathematics and Theoretical Physics, University of Cambridge, Cambridge, UK
[2]School of Earth and Environment, University of Leeds, Leeds, UK
[3]National Center for Atmospheric Research, Boulder, Colorado, USA

*Correspondence to:* Alison Ming (A.Ming@damtp.cam.ac.uk)

**Abstract.** The prominent annual cycle in temperatures (with maximum peak to peak amplitude of $\sim 8\,\mathrm{K}$ around $70\,\mathrm{hPa}$ and $\sim 6\,\mathrm{K}$ at $90\,\mathrm{hPa}$) is a key feature of the tropical tropopause layer (TTL). There is also a strong annual cycle observed in both ozone and water vapour in the TTL, with the latter understood as a consequence of the temperature annual cycle. The

radiative contributions of the annual cycle in ozone and water vapour to the temperature annual cycle are studied, first with a seasonally evolving fixed dynamical heating calculation (SEFDH) where the dynamical heating is assumed to be unaffected by the radiative heating. In this framework, the variations in ozone and water vapour derived from satellite data lead to variations in temperature that are respectively in phase and out of phase with the observed annual cycle. The ozone contribution is at the upper range of previous calculations. This difference in phasing can be understood from the fact that an increase in water vapour

cools the TTL, predominantly through enhanced local emission, whereas an increase in ozone warms the TTL, mostly through enhanced absorption of upwelling longwave radiation from the troposphere. The relative phasing of the water vapour and ozone effects on temperature is further influenced by the fact that for water vapour there is a strong non-local effect on temperatures from variations in concentrations occurring in lower layers of the TTL. In contrast, for ozone it is the local variations in concentration that have the strongest impact on local temperature variations. The factors that determine the vertical structure

of the annual cycle in temperature are also examined. Radiative damping time scales are shown to maximize over a broad layer centred on the cold point. Non-radiative processes in the upper troposphere are inferred to impose a strong constraint on temperature perturbations below $130\,\mathrm{hPa}$. These effects, combined with the annual cycles in dynamical and radiative heating, which both peak above the cold point, result in a maximum amplitude of temperature response that is relatively localized around $70\,\mathrm{hPa}$. Finally, the SEFDH assumption is relaxed by considering the temperature responses to ozone and water vapour

variations in a zonally symmetric dynamical model. While the magnitude of the tropical averaged temperature annual cycle in this framework is found to be consistent with the SEFDH results, the effects of the dynamical adjustment act to reduce the strong latitudinal gradients and inter-hemispheric asymmetry in the temperature response. This results in a temperature response that shows a considerably smoother structure than inferred from the SEFDH model. Whilst precise numerical values are likely to be sensitive to changes in the details of radiation code and of ozone and water vapour concentrations, the net

contribution to the annual cycle in temperature from both ozone and water vapour averaged between $20°\,\mathrm{N–S}$, calculated in this work, is substantial and around $35\,\%$ of the observed peak to peak amplitude at both $70\,\mathrm{hPa}$ and $90\,\mathrm{hPa}$.



## 1 Introduction

The tropical tropopause layer (TTL) (e.g., Fueglistaler et al. (2009)), in the pressure range 150 to 70 hPa (height range 14 to 18.5 km), is the main entry region for air into the stratosphere from the troposphere. The properties of this region are influenced by the presence of a prominent annual cycle in temperatures which is clear in, for example, radiosonde measurements (Reed and Vlcek (1969)) and GPS radio occultation measurements (Randel et al. (2003)). Fig. 1, shows the structure of the temperature annual cycle in a month-by-month climatology from the ERA-Interim reanalysis dataset (Dee et al. (2011)), constructed using data from 1991 to 2010. The structure is broadly consistent with that seen in earlier studies. The annual cycle temperature variations appear to be coherent over a layer from about 130 to 40 hPa, Fig. 1(a), with relatively cold temperatures in northern hemisphere (NH) winter and relatively warm temperatures in NH summer and early autumn, and with weak latitudinal gradients over the tropical (20° N–S) region, Fig. 1(b)–(d). The maximum amplitude occurs at around 70 hPa, with a 8 K peak to peak change, Fig. 1(b). At 90 hPa, the amplitude is about 6 K and at 100 hPa about 3 K, and the amplitude reduces very rapidly below 100 hPa (Fig. 1(a)). Above about 30 hPa (not shown), temperature variations are dominated by the semi-annual oscillation and whilst there is some evidence of variation on the annual frequency below 130 hPa, the amplitude is small.

Temperature variations at the tropical cold-point, at about 90 hPa, on annual and inter-annual time scales regulate the water vapour entering the stratosphere by freeze drying the upwelling air (e.g., Fueglistaler and Haynes (2005), Fueglistaler (2005), Randel and Jensen (2013)). The regular annual cycle at the cold-point, with colder temperatures in NH winter and warmer temperatures in NH summer, creates the well known water vapour tape recorder signal through this temperature-modulated freeze drying process followed by upward transport in the lower stratospheric Brewer-Dobson circulation (e.g., Mote et al. (1996), Randel et al. (2001)). According to Lagrangian trajectory calculations (e.g., Liu et al. (2010)), most of the freeze drying occurs in the 100 to 90 hPa region. Therefore Fig. 1(c) and (d) are more directly relevant to dehydration than the temperature at 70 hPa (Fig. 1(b)).

Despite the potential significance of the annual cycle in TTL temperatures, both in its role in determining stratospheric water vapour concentrations and also simply as a conspicuous and persistent aspect of temperature variation, the mechanisms responsible for the cycle are not yet completely clear. Furthermore, state-of-the-art chemistry climate models, e.g., within the CMIP5 dataset, still have large inter-model differences in the amplitude of the annual cycle at 100 hPa with peak to peak amplitudes ranging from $\sim 1\,$K to $\sim 5\,$K compared to $\sim 4$K for the 15° N–S average in ERA-Interim (Kim et al. (2013)).

In this paper we will examine further mechanisms for the annual cycle, with a particular emphasis on accounting for radiative heating in a realistic manner that goes beyond the Newtonian cooling approximation often made in theoretical investigations (e.g, Randel et al. (2002), Fueglistaler et al. (2014)). It is useful to begin by introducing the thermodynamic equation in the Transformed Eulerian Mean framework (Andrews et al. (1987)), neglecting eddy terms,

$$\partial_t \overline{T} = \overline{Q}_{\mathrm{rad}}(\overline{T}, \overline{\chi}_{\mathrm{O}_3}, \overline{\chi}_{\mathrm{H}_2\mathrm{O}}) - \overline{w}^* \overline{S} - \overline{v}^* \, \partial_y \overline{T} = \overline{Q}_{\mathrm{rad}}(\overline{T}, \overline{\chi}_{\mathrm{O}_3}, \overline{\chi}_{\mathrm{H}_2\mathrm{O}}) + \overline{Q}_{\mathrm{dyn}}, \qquad (1)$$

which predicts the rate of change of zonal mean temperature, $\overline{T}$, with time, $t$ where $\overline{(.)}$ represents a zonal mean. The dynamical heating, $\overline{Q}_{\mathrm{dyn}}$, is defined by the second equality in Eq. (1). $\overline{v}^*$ and $\overline{w}^*$ are the horizontal and vertical components of





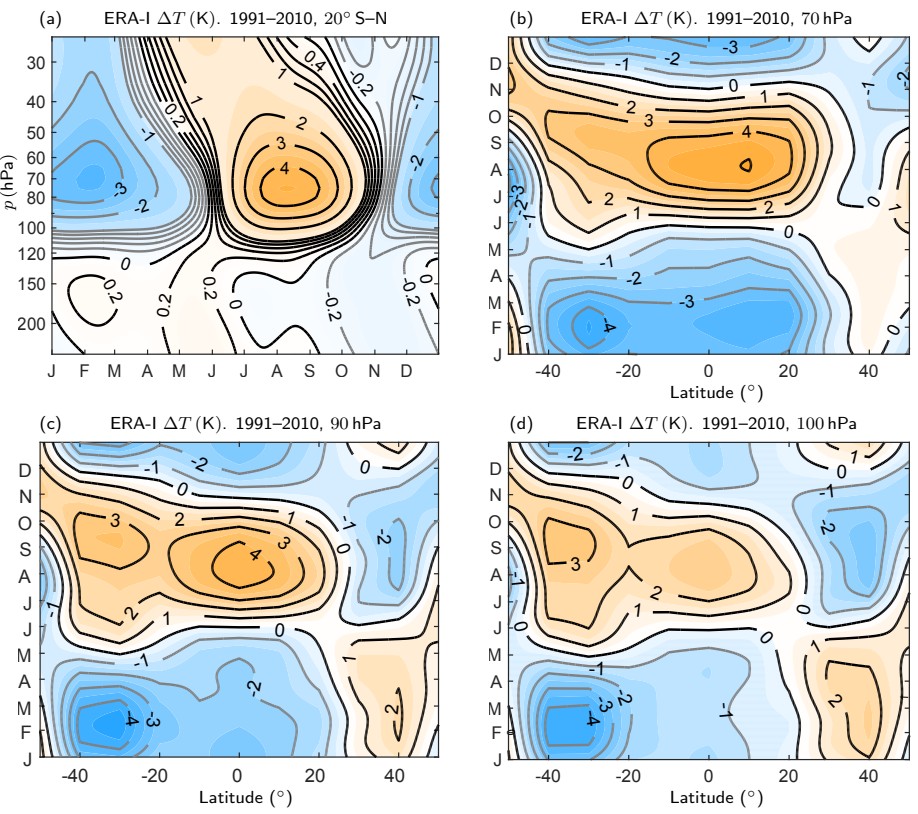

**Figure 1.** Climatology of annual cycle in ERA-Interim temperature constructed by averaging from 1991 to 2010 and (a) between $20°$ N–S, (b) at $70\,\mathrm{hPa}$, (c) at $90\,\mathrm{hPa}$ and (d) at $100\,\mathrm{hPa}$. The tick marks for the months indicate the first day of each month. The values of the fields are indicated by solid contours, with contour values labelled explicitly, and colours which change at intervals equal to half of the contour interval. These conventions are followed in all of the Figures in the paper.

the mean residual velocity respectively. $y$ is the meridional coordinate. $\overline{S} = \partial\overline{T}/\partial z + \kappa\overline{T}/H$ is a measure of the static stability where $z$ is the log-pressure height. $z = -H\log(\sigma)$ where $H$ is a scale height taken to be $7\,\mathrm{km}$ and $\sigma = p/p_0$ where $p$ is pressure and $p_0 = 1000\,\mathrm{hPa}$. $\kappa = R/c_p \simeq 2/7$ where $R$ is the gas constant for dry air and $c_p$ is the specific heat at constant pressure. The diabatic heating term on the right hand size, $\overline{Q}_{\mathrm{rad}}$, is the radiative heating and depends on the temperature and the distributions

5 of various radiatively active components including clouds, aerosols and trace gases. In the following, justified by the focus on the annual cycle in temperatures at $100\,\mathrm{hPa}$ and above, $\overline{Q}_{\mathrm{rad}}$, will be assumed to be the only component of diabatic heating and furthermore only the effect of variation in concentrations of ozone, $\overline{\chi}_{\mathrm{O_3}}$, and water vapour, $\overline{\chi}_{\mathrm{H_2O}}$, on $\overline{Q}_{\mathrm{rad}}$ will be taken into account.

Note that Eq. (1) goes along with three other coupled equations (zonal momentum, thermal wind and continuity equations)

10 to provide a complete set for the zonal mean temperature, zonal velocity and the vertical and latitudinal components of the





meridional circulation (Andrews et al. (1987), Chapter 3). This is particularly important in considering the response to an imposed forcing on the right-hand side of Eq. (1), or indeed an imposed forcing term in one of the other equations.

Many of the explanations proposed for the annual cycle in temperatures have focused on the role of dynamics. These emphasize the role of wave-induced forces in determining the strength of the Brewer-Dobson circulation and, in particular, the

upwelling velocity in the TTL. According to Eq. (1), this upwelling implies adiabatic cooling, i.e., negative values of $\overline{Q}_{\mathrm{dyn}}$, which drives temperatures away from their radiatively determined state. When upwelling is relatively strong, as is believed to be the case in NH winter relative to NH summer, temperatures are relatively cold. In the dynamical explanations the stronger upwelling in NH winter is caused by stronger wave forces during that season. A variety of waves have been identified as being important for this annual variation in tropical upwelling which include extratropical planetary waves, where the annual variation

might arise from inter-hemispheric differences in orography and other planetary-wave forcing (e.g., Yulaeva et al. (1994), Chen and Sun (2011), Ueyama and Wallace (2010), Ueyama et al. (2013)), tropical planetary waves (e.g., Kerr-Munslow and Norton (2006) and Ortland and Alexander (2014)), and baroclinic waves (e.g., Jucker et al. (2013)). Randel et al. (2008) and Grise and Thompson (2013) argue that all three are important.

However, radiative contributions to the annual cycle have also been suggested, principally in connection with the strong

annual cycle in TTL ozone concentrations (Folkins et al. (2006), Randel et al. (2007)). The quantitative effect of ozone on TTL temperatures was first investigated by Chae and Sherwood (2007) who used a one-dimensional radiative convective model representing a tropical average. They concluded that at $70\,\mathrm{hPa}$, about $3\,\mathrm{K}$ of the observed $8.2\,\mathrm{K}$ peak to peak variation in temperature might be caused by the radiative effects of the annual cycle in ozone, reducing to about $1\,\mathrm{K}$ of the observed $3\,\mathrm{K}$ peak to peak variation at $100\,\mathrm{hPa}$. Similar conclusions for $70\,\mathrm{hPa}$, though with a slightly smaller contribution of $2\,\mathrm{K}$ to

the peak to peak variation, were obtained by Fueglistaler et al. (2011) who used a Seasonally Evolving Fixed Dynamical Heating (SEFDH) framework. The SEFDH calculation (see Sect. 3) considered the response of Eq. (1) to a change in ozone concentrations from annual mean to annual varying values, assuming that the dynamical heating is unaffected by the change.

Alongside ozone, the other radiatively active species which has a strong annual cycle in the TTL region is water vapour, as mentioned above. Both Chae and Sherwood (2007) and Fueglistaler et al. (2011) asserted that the role of annual variation in

water vapour concentrations in determining the annual cycle in temperatures was small; however, neither paper gives quantitative details. There has been recent significant interest in the radiative effect of variations in stratospheric water vapour, both the effect on the radiative balance of the troposphere (Dessler et al. (2013), Maycock et al. (2013), Joshi et al. (2006), Forster (2002), Solomon et al. (2010)), but also the effect on the lower stratosphere. For example, Maycock et al. (2011), used a set of radiative calculations to show that a uniform increase in stratospheric water vapour gives rise to a cooling that is largest in the

lower stratosphere at all latitudes.

In this work, we describe an investigation, first based on the SEFDH framework, of the individual and combined effects of the annual cycles in ozone and water vapour on TTL temperatures. Since in the real atmosphere and in models, different processes are likely to contribute to the variation at different vertical levels, we include details of the contributions from different vertical regions. The radiative calculations required for this investigation also allow us to examine carefully how different aspects of

the dynamical state and the temperature- and species-dependence of the radiative heating determine the vertical structure of the





annual cycle in temperatures including the amplitude maximum at 70 hPa and the amplitude at 90 hPa, which plays a crucial role in determining stratospheric water vapour concentrations.

For a complete assessment of the effect of seasonal variations of ozone and water vapour on the annual cycle in TTL temperatures it is necessary to take account of dynamical changes induced by these variations. For example, a change in ozone

in a zonally symmetric atmosphere will change heating rates leading to a direct change in the meridional circulation and hence in $\overline{w}^*$. This is simply the 2D zonally symmetric dynamical response to change in heating, as described in many previous papers such as Plumb (1982), Garcia (1987) and Haynes et al. (1991). Temperature changes will also contribute to changes in the static stability and affect the dynamical heating term. Additionally, in a 3D dimensional atmosphere, part of the response to the change in ozone heating (even if the latter is independent of longitude) will be a change in the wave-induced force because

the propagation and dissipation characteristics of the waves will be changed (see e.g., Ming et al. (2016)). This too will lead to a change in the meridional circulation and hence in $\overline{w}^*$. Both of these distinct contributions to a change in $\overline{w}^*$ will modify the change in temperature from that predicted by the SEFDH approach, which assumes that the dynamical heating is fixed. Therefore, this potentially has important implications for the temperature response to variations in ozone and water vapour and we also assess this contribution here.

The structure of the paper is as follows. Section 2 describes the data and the radiative calculations. Section 3 describes the SEFDH calculations to quantify the effect of seasonal variations in ozone and water vapour on the annual cycle of temperature in the TTL. This Section includes some detailed discussion of the radiative effect of variations in water vapour, neglected by previous authors, and of the non-local interactions in the vertical that determine how the vertical structure of the variations in ozone and water vapour relate to the vertical structure of the corresponding variations in temperature. This Section is

complemented by a set of illustrative FDH radiative calculations which are included in Appendix A. Section 4 discusses the vertical structure of the annual cycle in temperature, analysing the effects of the vertical structure of the background temperature field on the relaxational part of the heating, of ozone and water vapour, and of dynamical quantities, in particular the upwelling velocity and the static stability. More details about the estimates of uncertainty associated with the calculations in Sect. 3 and 4 are given in Appendix B. Section 5 then goes beyond the SEFDH calculation reported in Sect. 3 to consider how the

temperature response to variations in ozone and water vapour is affected by the change in dynamical heating captured by a 2D zonally symmetric dynamical response calculation. A final Section discusses the results and their implications and reviews the various simplifying assumptions that have been made.

## 2 Data and Radiative method

We make use of the temperature and dynamical data from the ERA-Interim dataset covering the period 1991 to 2010, using

data at a horizontal resolution of $1°$, at 6-hourly analysis time intervals (0000, 0006, 0012 and 0018 UTC) and on 60 model levels. The mean residual vertical velocity in the Transformed Eulerian Mean framework, calculated using the same method as Seviour et al. (2012), and the dynamical heating used in Sect. 4 are both computed on the original grid from the ERA-





Interim data and then smoothed by linearly interpolating the monthly averages to daily values. The temperature is also linearly interpolated to the grids relevant for the calculations described below and in Sect. 5.

Ozone and water vapour mixing ratios are obtained from the Stratospheric Water and OzOne Satellite Homogenized (SWOOSH) dataset (Davis et al. (2016), Tummon et al. (2015)). This record is formed from a combination of measurements from various

limb and solar occultation satellites from 1984 to 2015, namely: SAGE-II/III, UARS HALOE, UARS MLS, and Aura MLS instruments. The measurements are homogenized by applying corrections that are calculated from data taken during time periods of instrument overlap. Although this dataset is relatively new, it has been used in a study of stratospheric water vapour variability (Maycock et al. (2014)) and makes uses of HALOE and Aura MLS which have been widely validated in the literature (Harries et al. (1996), Bruhl et al. (1996), Lambert et al. (2007)). SWOOSH is chosen for this study because it provides a

homogenized record useful for climate studies. The lowest pressure level in SWOOSH is $316\,hPa$. The results presented in this paper are not sensitive to concentrations of water vapour and ozone below $316\,hPa$ (if these are within reasonable limits) and for convenience the profiles below $316\,hPa$ were simply defined by linear interpolation between the SWOOSH value at $316\,hPa$ and the surface values taken from ERA-Interim.

The radiative calculations were performed using a modified version of the Morcrette (1991) radiation scheme, which includes

updates to the longwave absorption properties of water vapour (Zhong and Haigh, 1995). All calculations were done on zonal mean data at $5°$ intervals in latitude and on 100 pressure levels (which are the same as those listed in Appendix A for the FDH calculations).

In the shortwave, a three point Gaussian quadrature method is used to account for the diurnal variation in the solar zenith angle. The albedo is taken from ERA-Interim reanalysis data. Carbon dioxide is assumed to be well mixed and the volume

mixing ratio is set to $360\,ppmv$.

To study the radiative contributions of seasonal variations in ozone and water vapour to the annual cycle in TTL temperatures, we make use of the Seasonally Evolving Fixed Dynamical Heating calculation (SEFDH) (Forster et al. (1997)). This method calculates the time varying temperature change due to a specified time varying trace gas perturbation and takes into account the specified time dependence of a background state with temperature $T_0(z,t)$ and species concentration $\chi_0(z,t)$ to which the

trace gas perturbation is applied.

Given time-varying background profiles (at specified latitude) of temperatures and concentrations of trace gases ($\overline{T}^0$, $\overline{\chi}^0_{O_3}$, $\overline{\chi}^0_{H_2O}$, where $(\cdot)^0$ denotes the background state), the dynamical heating, $\overline{Q}^0_{dyn}$, is first calculated by assuming the balance in Eq. (1), i.e.,

$$\partial_t \overline{T}^0 = \overline{Q}_{rad}(\overline{T}^0, \overline{\chi}^0_{O_3}, \overline{\chi}^0_{H_2O}) + \overline{Q}^0_{dyn}. \tag{2}$$

A perturbation is applied to trace gas concentrations ($\Delta\overline{\chi}_{O_3}, \Delta\overline{\chi}_{H_2O}$) and the new time evolving equilibrium temperature state, $\overline{T}^0 + \Delta\overline{T}$, is obtained from

$$\partial_t(\overline{T}^0 + \Delta\overline{T}) = \overline{Q}_{rad}(\overline{T}^0 + \Delta\overline{T}, \overline{\chi}^0_{O_3} + \Delta\overline{\chi}_{O_3}, \overline{\chi}^0_{H_2O} + \Delta\overline{\chi}_{H_2O}) + \overline{Q}^0_{dyn}. \tag{3}$$





Note that all quantities are shown as zonal averages for consistency. It is necessarily the case that the temperature change will be zonally uniform since the trace gas contributions are zonal averages. By definition, each calculation is local in latitude. The specification of the time dependent radiative heating includes the time variation of the zenith angle.

To generate the results reported in the following Sections we take the background state to be the annual average ERA-
Interim temperature and the annual mean SWOOSH constituent concentration. The latter are then perturbed to the annual cycle concentration. The quantitative impact of perturbing around a different background state which corresponds to the annual cycle in temperature (the method used by Fueglistaler et al. (2011)) on the predicted temperature response was very small indeed (0.09 K at most, in the tropical lower stratosphere) and taking the background temperature to be time independent is easier to implement in the dynamical calculations reported in Sect. 5.

Following a similar method to Forster et al. (1997), Eq. (3) is applied to update the temperature only above a certain level which we choose, given our focus on the tropics, to be 130 hPa. The variation of the various trace gases is not fixed below this level and can have a radiative effect on the temperatures above 130 hPa and this effect will be quantified in Sect. 3. The rationale for applying Eq. (3) only above 130 hPa is that, as noted by Fueglistaler et al. (2009b), the processes determining temperature variations in the tropical troposphere below the TTL, where convective effects are strong, are distinct from those
determining variations in the TTL. Some kind of ad hoc modification to the radiative calculation, e.g., by reducing radiative time scales to represent radiative-convective adjustment, could be implemented, but since it is not the variation of temperature below 130 hPa that is of interest in this study, but rather any effect of those temperatures on the TTL temperatures, we simply clamp the temperatures below 130 hPa so that the annual variation is zero. This is a simple representation of what is observed (see Fig. 1(a)) and indeed it is the abrupt decrease in the amplitude of the observed annual cycle below 130 hPa that motivates
this approach.

In setting up the calculation we verified as a test that the Forster et al. (1997) results could be reproduced. In our application of the method the basic state temperature and species concentrations and the perturbation species concentrations are annually repeating and Eq. (3) is therefore integrated forward in time with a daily time step until the perturbed temperature field is also annually repeating. This integration is carried out to five years which is sufficient for accurate convergence to the annually re-
peating state. The SEFDH technique reduces to the more standard and widely used Fixed Dynamical Heating (FDH) technique if the imposed background temperature and species fields and the perturbation concentrations in species are constant in time and Eq. (3) is integrated to a steady state (Appendix A).

## 3 Seasonally Evolving Fixed Dynamical Heating (SEFDH) calculations

### 3.1 Ozone annual cycle

Figure 2 shows the difference in the ozone mixing ratio from the annual mean over the tropical region. The annual cycle in lower stratospheric ozone concentrations, and in particular the large amplitude of the annual cycle relative to annual mean values in the tropical lower stratosphere, is well known on the basis of sonde data (e.g., Logan (1999)) and data from satellite instruments such as HALOE and MLS (e.g., Randel et al. (2007), Tegtmeier et al. (2013)). The height and latitude structure





of the annual cycle at low latitudes, as represented in the SWOOSH dataset, is shown in Fig. 2(a), which shows the difference from the annual mean in the ozone concentration, averaged over the tropical region, as a function of height and Fig. 2(b), which shows the corresponding latitudinal structure at 70 hPa. Ozone concentrations are lowest across the tropics in NH winter/spring and highest in NH summer/autumn. The cycle has a broad latitudinal structure, but the amplitude is significantly larger in the

NH subtropics than in the SH subtropics. Whilst the amplitude of the annual cycle in ozone concentration increases with height (Fig. 2(a)), the amplitude as a proportion of the annual mean concentration (the 'relative amplitude') is largest at about 80 hPa and decreases upward above that level (e.g., Randel et al. (2007), their Fig. 3).

Randel et al. (2007) note the narrow vertical structure of the relative amplitude in ozone, so that, for example, HALOE data shows a strong maximum whereas MLS data, which has relatively coarser vertical resolution, shows a much weaker maximum.

They argue that the vertical structure will only be properly resolved by measurements with high vertical resolution, such as those provided by sonde measurements. In using the SWOOSH dataset, which provides ozone values at 68.1, 82.5 and 100 hPa in this region, we therefore need to accept that the maximum may not be well resolved. It is important to emphasize that the total amplitude of the change in trace gas, as well as the background concentration, are what determine the temperature change and not the relative amplitude of the trace gas change.

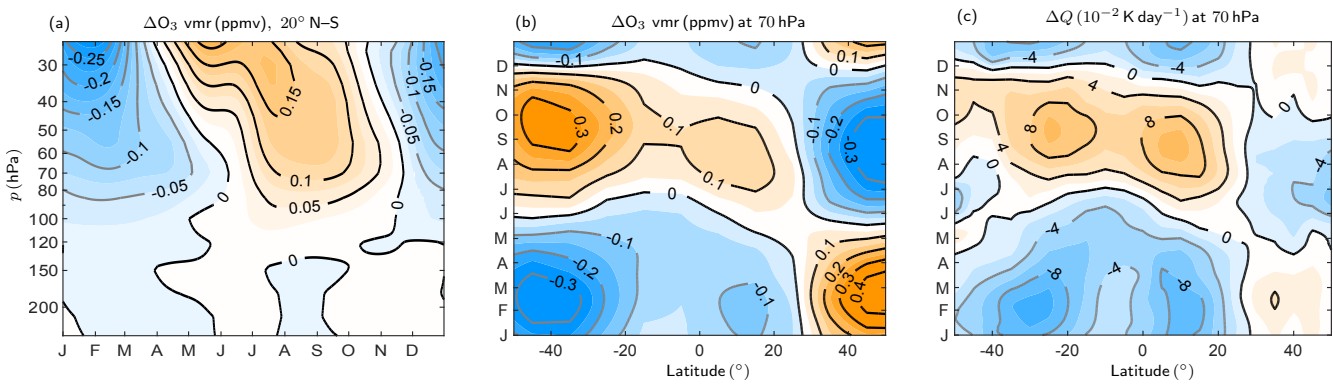

**Figure 2.** Ozone volume mixing ratio (ppmv) from the SWOOSH dataset plotted as a difference from the annual mean (a) averaged between $20°$ N–S and (b) at 70 hPa. (c) The total change in heating rate (longwave and shortwave) $(10^{-2}\,\mathrm{K\,day^{-1}})$ due to the ozone annual cycle.

The mechanisms for the annual cycle in ozone have been discussed in the recent literature, with differing emphasis on the role of vertical transport and horizontal transport from the extratropics. Abalos et al. (2013) summarise the recent work and conclude that both vertical and horizontal transport are important, with the latter causing significant variation in ozone concentrations in the lower part of the TTL and the former communicating this variation in the vertical. The latitudinal structure has also been discussed in Stolarski et al. (2014) who argued that the ozone annual cycle in the southern tropics is dominated

by the seasonality of upwelling whereas in the northern tropics, seasonality in mixing is more important. Inter-hemispheric differences in the magnitude and phase of the variations in upwelling and mixing contribute to the observed structure.





The SEFDH procedure described in Sect. 2 (see also Appendix A) was applied to determine the radiative effects of the annual cycle in ozone. The predicted temperature variation is shown in Fig. 3(a) and (b). A detailed explanation of the temperature response to a change in ozone concentration is given in Appendix A1, but broadly speaking, the main effect of an increase in ozone in a particular shallow layer of the TTL or lower stratosphere is to increase heating in that layer, through both increased

shortwave absorption and increased absorption of upwelling longwave radiation. At these pressure levels, changes due to ozone in longwave heating dominate over shortwave changes. The increased longwave absorption acts to reduce heating above that layer. As an example, Fig. 2(c) shows the change in heating rate at $70\,\mathrm{hPa}$ due to the ozone annual cycle. (The quantity plotted is $\overline{Q}_{\mathrm{rad}}(\overline{T}^0, \overline{\chi}^0_{\mathrm{O_3}} + \Delta\overline{\chi}_{\mathrm{O_3}}, \overline{\chi}^0_{\mathrm{H_2O}}) - \overline{Q}_{\mathrm{rad}}(\overline{T}^0, \overline{\chi}^0_{\mathrm{O_3}}, \overline{\chi}^0_{\mathrm{H_2O}})$, where, as set out previously, $(\cdot)^0$ here denotes the annual mean value for each quantity and $\Delta\overline{\chi}^0_{\mathrm{O_3}}$ is the annual cycle variation in ozone concentration.) At this level the peak to peak change in longwave

heating at $70\,\mathrm{hPa}$ between $20°\,\mathrm{N}$–$\mathrm{S}$ is about $0.11\,\mathrm{K\,day^{-1}}$ while the shortwave heating is about $0.03\,\mathrm{K\,day^{-1}}$. The increased longwave absorption acts to reduce heating above that layer. Consistent with these arguments, the SEFDH calculation predicts a significant annual cycle in temperature across the tropics (averaged between $20°\,\mathrm{N}$-$\mathrm{S}$) caused by the variation in ozone, with cooler temperatures when ozone concentrations are relatively low, in NH winter, and warmer temperatures when ozone concentrations are relatively high, in NH summer.

From the vertical structure of the amplitude of the temperature annual cycle shown in Fig. 3(a) it can be seen that the amplitude is largest between 70 to $90\,\mathrm{hPa}$ with a peak to peak amplitude of about $3.5 \pm 0.4\,\mathrm{K}$ at $70\,\mathrm{hPa}$ and about $3.3 \pm 0.5\,\mathrm{K}$ at $90\,\mathrm{hPa}$ (where the values are quoted with 95% confidence intervals that are calculated from the uncertainty obtained from the SWOOSH dataset, see Appendix B for more details). The temperature annual cycle has a lag of about 1.5 months compared to the annual cycle in ozone (e.g., the minimum ozone is in February whereas the minimum in the temperature response is in

March-April). This lag is consistent with radiative time scales in this region of the atmosphere (see discussion in Sect. 4). Note that the temperature response essentially has the same sign at all levels, presumably because the change in ozone concentrations occurs over a relatively deep layer so that, at a given level, any effects of the reduction in upwelling radiation by increased ozone in the levels below are dominated by the increased absorption at that level.

The SEFDH calculation also predicts the latitudinal structure of the temperature response which is shown in Fig. 3(b). Note

that the SEFDH calculation is local in latitude, so that temperature variation at a given latitude is determined solely by ozone variation at the same latitude. The temperature response at each latitude is again seen to lag the ozone variation by about 1.5 months. There is generally a good match between the latitudinal structure in the amplitude of the temperature response and that of the ozone variation shown in Fig. 2(b), in particular, both are relatively strong in the NH subtropics and relatively weak in the SH subtropics. As we move south of the Equator, the change in ozone is a local minimum in February at $40°\,\mathrm{S}$, but the

temperature change is a minimum at around $30°\,\mathrm{S}$. This is due to the upwelling longwave radiation decreasing as the opacity of the underlying atmosphere increases.

The temperature variation we obtain at $70\,\mathrm{hPa}$ has a broadly similar structure in latitude and time to that presented by Fueglistaler et al. (2011) who used ozone from the HALOE dataset and the Edwards and Slingo radiation code. However the peak to peak amplitude we obtain, $3.5 \pm 0.4\,\mathrm{K}$, is significantly larger than the $2\,\mathrm{K}$ peak to peak amplitude obtained by

Fueglistaler et al. (2011). Chae and Sherwood (2007) used a simple radiative-convective model and found that ozone radiative



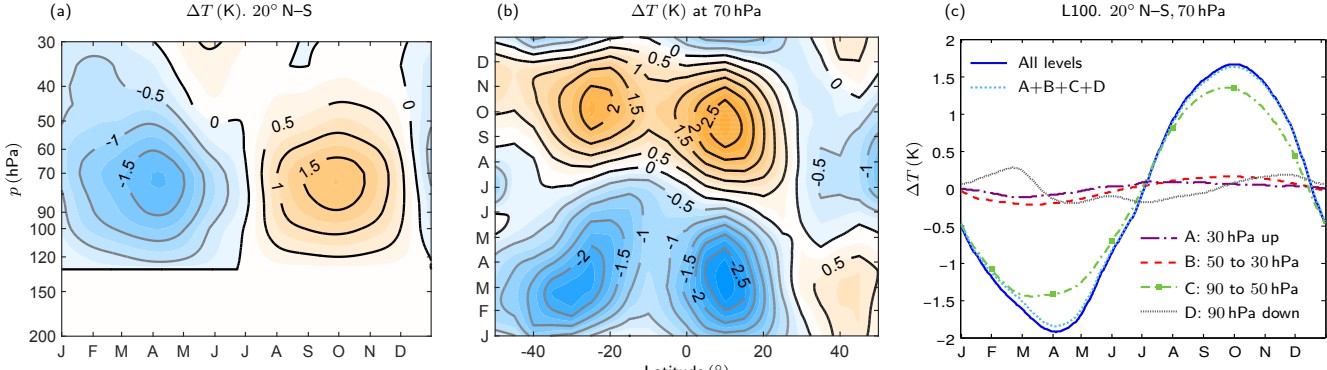

**Figure 3.** SEFDH temperature change (K) due to the annual cycle in ozone. All other trace gases are kept at their annual mean values. (a) Temperature change averaged between $20°$ N–S. (b) Temperature change at $70\,\text{hPa}$. (c) Temperature changes at $70\,\text{hPa}$ and averaged between $20°$ N–S calculated with the annual cycle in ozone imposed within different pressure ranges. Outside of each range and including the pressure level at the lower bound (in terms of height) of the range, the ozone mixing ratio is kept at the annual mean value.

effects contributed about $3\,\text{K}$ to the peak to peak temperature annual cycle at $70\,\text{hPa}$. Both our estimate of the contribution from ozone radiative effects and that of Chae and Sherwood (2007) are therefore larger than found by Fueglistaler et al. (2011).

There are several possible causes for the quantitative discrepancy of around a factor of about 1.8 between the results for the effect of the ozone annual cycle on temperatures shown in Fig. 3 and those of Fueglistaler et al. (2011). These possible

causes include: differences in the satellite ozone datasets employed, including both mean ozone concentration and annual cycle variations; and/or differences in radiation scheme performance. This study uses the SWOOSH dataset, which covers 1984 to 2015, whereas Fueglistaler et al. (2011) used filled HALOE data covering the period 1994 to 2000. While the amplitude of the mean annual cycle is not expected to have changed substantially between these time periods, the use of different ozone datasets may contribute to the differences identified above. Fueglistaler et al. (2011) show fractional ozone changes over the annual

cycle in their Fig. 5(a), but without knowledge of the mean ozone value to which their annual cycle anomalies correspond, it is difficult to make a direct comparison with the ozone annual cycle derived from SWOOSH shown in Fig. 2(b). What is clear is that the quantitative response to the ozone annual cycle is extremely sensitive to the mean background ozone value. For example, a change in the background ozone concentration of $\pm 0.05\,\text{ppmv}$ (from $0.5\,\text{ppmv}$) around $70\,\text{hPa}$ (the uncertainty in ozone mixing ratio from the SWOOSH dataset) leads to a change of $\pm 0.4\,\text{K}$ in the peak to peak temperature amplitude

at that level (see Sect. 4 for plots showing the uncertainty and Appendix B for further details). Thus the use of different ozone datasets may contribute to the differences between our study and Fueglistaler et al. (2011). Another factor that may be important is the choice of radiation code used to perform the SEFDH calculations. Broadband radiation schemes may have different intrinsic sensitivities to absorbers as a result of the methods adopted to parameterize the radiative transfer equation. The stratospheric temperature response to ozone perturbations in the modified Morcrette/Zhong and Haigh scheme used here

has been shown to agree closely with a more accurate Narrow Band Model (Forster et al. (2001)). Fueglistaler et al. (2011)





used the Edwards and Slingo radiative model, which has also been evaluated in relation to its performance for ozone in the troposphere and stratosphere, although we are not aware of a publication that directly compares the Edwards and Slingo and the modified Morcrette/Zhong and Haigh models for ozone. Nevertheless, given these prior model evaluations it seems unlikely that differences between the radiative codes in question could alone account for a factor of about 1.8 difference in the SEFDH

temperature response to the ozone annual cycle. It is also pertinent to mention here that we are able to quantitatively reproduce the SEFDH results of Forster et al. (1997) for an ozone perturbation despite the fact that different radiation codes were used. It therefore seems most likely that differences between the input ozone datasets are responsible, not withstanding the possibility for other unidentified methodological differences to play a role.

In order to investigate the role of ozone variations at different levels in determining the temperature variation, the SEFDH

calculation was repeated with the annual cycle in ozone imposed only within a range of pressure levels, namely: $[1, 30]$ hPa, $[30, 50]$ hPa, $[50, 90]$ hPa, and $[90, 1000]$ hPa (where $[\cdot]$ indicates a range of levels). Outside the chosen range and including the pressure level at the lower bound of the range (in height), ozone concentrations are set to the annual mean value. Figure 3(c) shows that the annual cycle in ozone in the region 90 to 50 hPa accounts for about 80% of the temperature annual cycle at 70 hPa, i.e., at this level the temperature variation is driven primarily by local variations in ozone. This was also found to

be true at 90 hPa (results not displayed), where about 60% of the temperature variation is driven by ozone variation in the $[80, 100]$ hPa layer and about 30 % by that in the $[50, 80]$ hPa layer. The temperature variations in the TTL region due to ozone variations are therefore driven primarily by local variations (perhaps extending up to about 50 hPa).

To summarise the results of this subsection, we have shown using an SEFDH calculation that the ozone annual cycle may account for $3.5 \pm 0.4$ K of the $8.2 \pm 0.3$ K observed peak to peak amplitude of the annual cycle in temperature at 70 hPa averaged

between $20°$ N-S, $3.3 \pm 0.5$ K of the $5.9 \pm 0.2$ K at observed at 90 hPa and $2.6 \pm 0.2$ K of the $3.4 \pm 0.1$ K at observed at 100 hPa. Whilst the precise numbers may be uncertain (e.g., recall the Fueglistaler et al. (2011) results for 70 hPa) the conclusion is that the radiative effect of ozone variations makes a substantial contribution to the observed annual cycle in temperature.

### 3.2   Annual cycle in water vapour

In order to determine the radiative effect of variations in water vapour on temperatures, we now repeat the SEFDH calculations

with water vapour rather than ozone as the varying species. Figure 4 shows the annual cycle in water vapour as represented in the SWOOSH dataset. The vertical structure of the time varying tropical average water vapour concentrations is shown in Fig. 4(a) (as anomalies with respect to the annual mean), with the tape recorder signal clearly visible as tilted bands of positive and negative anomalies. The amplitude of the stratospheric annual cycle is largest at around 90 hPa where temperatures are coldest, consistent with the fact that water vapour concentrations are directly modulated by the cold-point temperatures. Note that the

amplitude significantly increases downward below 150 hPa where the warmer temperatures imply that even small temperature variations can lead to large variations in water vapour concentrations relative to stratospheric values and additionally, of course, water vapour is likely to be significantly affected by convective processes.

Since the vertical structure of the water vapour annual cycle is quite complicated relative to that of ozone, we show the latitudinal structure at several different levels, 70, 90 and 100 hPa, Fig. 4(b)–(d) respectively. Note some hemispheric differences





are apparent, and that these are more pronounced at 100 hPa compared to 70 hPa. The maximum amplitude in the annual cycle in water vapour tends to be shifted towards the NH, perhaps because the annual cycle in the NH subtropics is enhanced by the relatively high values in summer. These are generally agreed to arise from the presence of the Asian, and to some extent the American, monsoon circulations (e.g., Rosenlof et al. (1997), Ploeger et al. (2013), Randel et al. (2015)) though the pre-

5   cise mechanism (organisation of the large-scale circulation and temperature field versus isolation of upward transported air) remains a subject of discussion.

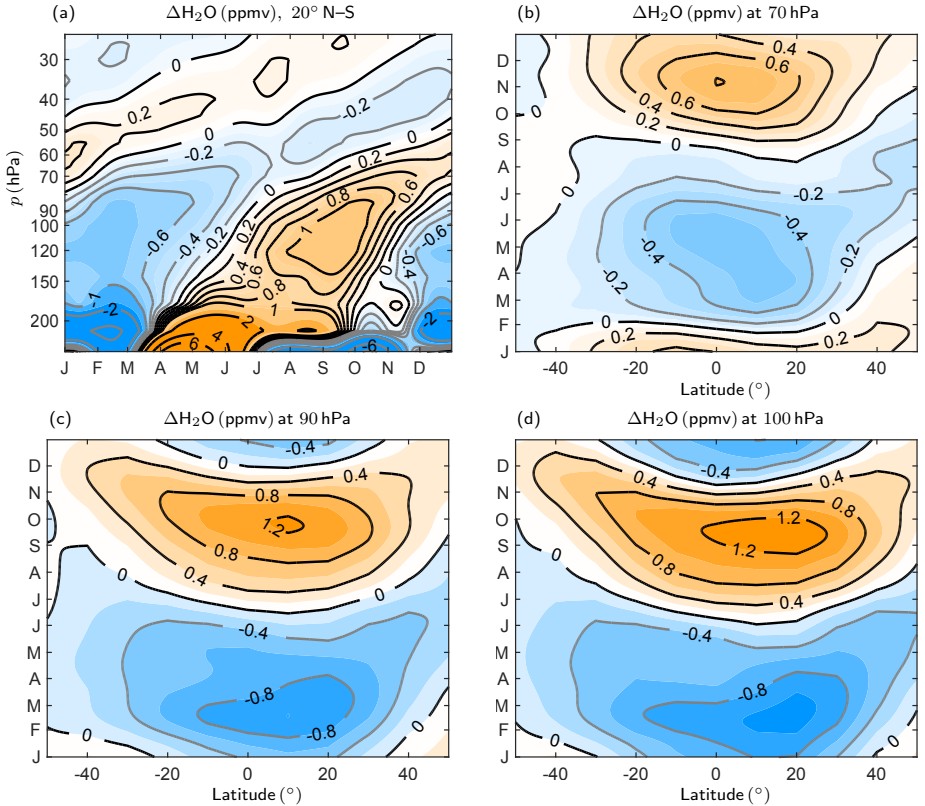

**Figure 4.** Water vapour volume mass mixing ratio (ppmv) from SWOOSH plotted as a difference from the annual mean (a) averaged over the region $20°$ N–S, (b) at 70 hPa, (c) at 90 hPa and (d) at 100 hPa.

Figure 5(a) shows that the SEFDH temperature change due to water vapour peaks at around 90 hPa, i.e., in the cold-point region and lower down than the peak change due to ozone, shown in Fig. 3(a). In contrast to the change due to ozone, where the phase of the resulting annual cycle in temperatures is essentially the same at all heights in the range 50 to 100 hPa, the

10   temperature change due to water vapour has a phase lag of about one month between the annual cycles at 90 hPa and at 70 hPa. Note that there is a phase lag between annual cycle in water vapour concentrations between these levels (Fig. 4(a)) of about 2 months. Notwithstanding this phase variation, the temperature annual cycle in the range 100 hPa to 70 hPa is, broadly speaking,




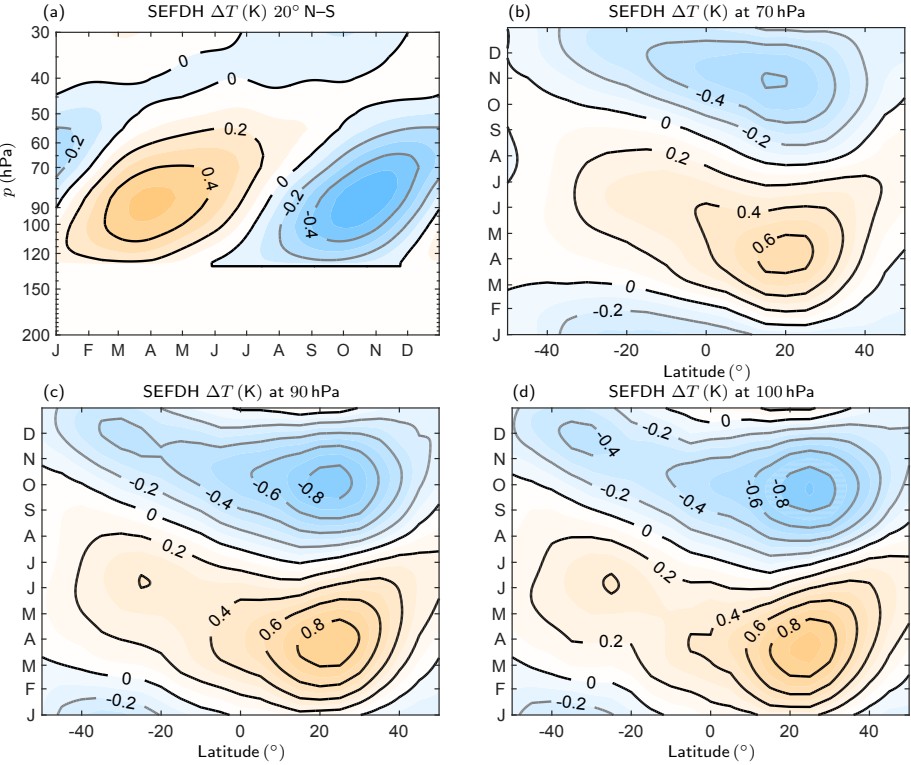

**Figure 5.** Temperature change (K) due to the water vapour annual cycle in an SEFDH calculation (a) averaged between $20°$ N–S, (b) at $70\,\text{hPa}$, (c) at $90\,\text{hPa}$ and (d) at $100\,\text{hPa}$.

opposite in phase to the observed annual cycle in temperatures (Fig. 1(a)) and the SEFDH-predicted annual cycle due to ozone (Fig. 3(a)).

Figures 5(b)–(d) show that the amplitude of the SEFDH temperature change from water vapour is largest in the NH at all three levels 70, 90 and $100\,\text{hPa}$. At each level, the latitude of the maximum amplitude in temperature change is located further

5    north than the latitude of the maximum amplitude in the water vapour annual cycle at that level. The fact that there is no simple relation between the latitude-time structure of the SEFDH-predicted annual cycle in temperature at a given level and the latitude-time structure of the water vapour concentration at that level, together with fact evident from Fig. 5(a) and noted above, that the phase of the temperature response at $70\,\text{hPa}$, say, is closer to the phase of the water vapour concentration at $90\,\text{hPa}$, rather than that at $70\,\text{hPa}$, suggests that there are important non-local contributions from changes in water vapour to the

10   temperature.

This non-local contribution of water vapour to the temperature response may be examined using the SEFDH approach, by imposing the annual cycle in water vapour only in limited pressure ranges: $[1, 60]\,\text{hPa}$, $[60, 80]\,\text{hPa}$, $[80, 100]\,\text{hPa}$, $[100, 130]\,\text{hPa}$, $[130, 200]\,\text{hPa}$, and $[200, 1000]\,\text{hPa}$. Typical results are illustrated by Fig. 6(a), which shows the temperature change at $90\,\text{hPa}$





for each of these calculations. The total peak to peak temperature change from water vapour at $90\,\mathrm{hPa}$ is $1.1 \pm 0.1\,\mathrm{K}$ which can be broken down as follows. There is a large local contribution of $0.7\,\mathrm{K}$ from $[80, 100]\,\mathrm{hPa}$, and a significant contribution of $0.4\,\mathrm{K}$ from the region $[100, 130]\,\mathrm{hPa}$. Contributions from above $80\,\mathrm{hPa}$ and from below $130\,\mathrm{hPa}$ are small. Note that the contribution from constituent changes from the $[130, 200]\,\mathrm{hPa}$ to the temperature change appears small in the $20°\,\mathrm{N–S}$ average but the latter has a large meridional gradient. It is larger in the Northern Hemisphere than the Southern Hemisphere (and out of phase to it) and contributes to about 15% of the temperature change at $90\,\mathrm{hPa}$ and $20°\,\mathrm{N}$ and increases outside the tropics (not shown).

Further illustration is provided by Fig. 6(b) which shows the time evolution of the temperature change at all pressure levels when the water vapour perturbation is confined to $[100, 130]\,\mathrm{hPa}$. Note that the water vapour in this layer is at a minimum in February-March and at a maximum in September-October. The response to reduced water vapour in a given layer in the TTL and lower stratosphere is cooling below that layer and heating within and above that layer. Increased water vapour in the layer gives the opposite response. For a detailed analysis of this response see Appendix A2. A brief explanation is that primarily the reduction in water vapour implies less local emission and therefore reduced absorption above and below, together with less absorption of upwelling radiation within the layer and increased absorption above. Within the layer the effect of reduced emission is stronger, so that there is heating. Above the layer the increased absorption of upwelling radiation is stronger and hence heating. Below the layer the only effect is reduced absorption and there is therefore cooling (although the temperatures are not allowed to change in this region). A secondary aspect of the behaviour is a small phase lag of the response above the layer relative to the water vapour change, since the effect is being communicated in part by changes in temperature, and hence changes in radiation, in intermediate layers. These features are all visible in the response shown in Fig. 6(b), except that there is no cooling below $130\,\mathrm{hPa}$ because, as part of the SEFDH implementation, the change in temperature is constrained to be zero there. (A similar description seems to hold for the response to changes in water vapour in different layers in the TTL and lower stratosphere). Figure. 6(c) shows the temperature change at $90\,\mathrm{hPa}$ from water vapour changes in the $[100, 130]\,\mathrm{hPa}$ region. Comparing to Fig. 5(c) (note different contour interval), the $[100, 130]\,\mathrm{hPa}$ region contributes to about 35 % of the temperature change at $90\,\mathrm{hPa}$. Figure. 6(c) also shows that the peak in amplitude in the temperature change from the $[100, 130]\,\mathrm{hPa}$ region is centred about $25°\,\mathrm{N}$ and this supports the idea that water vapour lower down than $90\,\mathrm{hPa}$ shifts the peak in the latitudinal structure of total temperature change at $90\,\mathrm{hPa}$ (Fig. 5(c)) northwards from what it would be from only the local water vapour.

### 3.3 Annual cycle in both constituents and dynamical heating

Figure 7(a), for $70\,\mathrm{hPa}$, and Fig. 7(b), for $90\,\mathrm{hPa}$, show the combined temperature effect (solid grey curves with dot markers) of imposing the annual cycles in ozone and water vapour together in an SEFDH calculation. These figures also show the observed annual cycle in temperature (solid green curves), the estimated annual cycle in temperature due to dynamical heating (dashed light blue curve) where the dynamical heating has been calculated from ERA-Interim data (see Sect. 4 for further details on how the dynamical heating is added as a forcing) and the estimated annual cycle due to the combined effect of ozone, water vapour and dynamical heating (solid red curves with square markers). To a good approximation the effects of ozone and water vapour add linearly, so the combined effect is simply the sum of the individual effects discussed in Sect. 3.1 and 3.2.



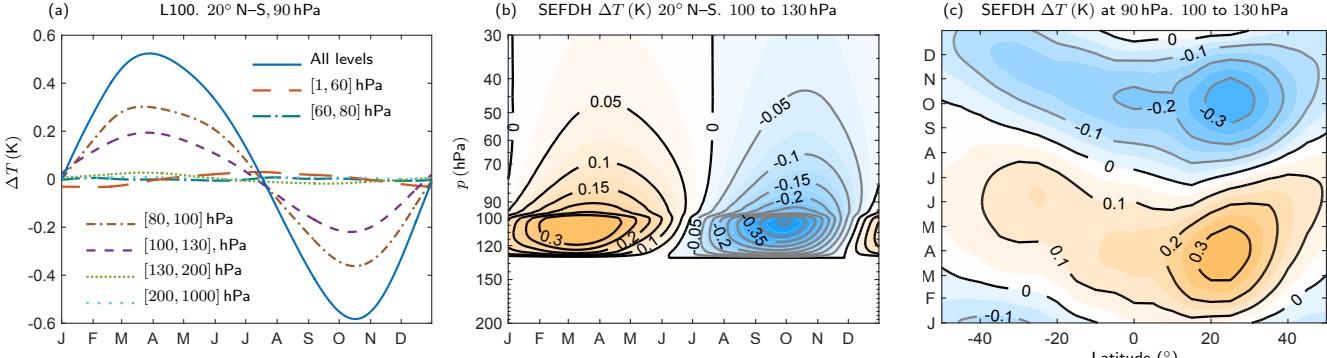

**Figure 6.** Annual cycle temperature changes (K) at $90\,\mathrm{hPa}$ calculated using SEFDH with the annual cycle in water vapour imposed within different pressure ranges. Outside of this range and on the pressure level at the lower bound (in terms of height) of the range, the water vapour mixing ratio is kept at the annual mean value. The plots are averaged between $20°\,\mathrm{N–S}$. The contributions from each layer add linearly to reproduce the total change (not shown). The temperature change (K) for the case in (a) where the water vapour annual cycle is imposed only in the range $[100, 130]\,\mathrm{hPa}$ is shown in (b) averaged between $20°\,\mathrm{N–S}$ and in (c) at $90\,\mathrm{hPa}$.

At $70\,\mathrm{hPa}$ (Fig. 7(a)), the combined effect of both ozone and water vapour is an annual cycle in temperature of about $3.1 \pm 0.3\,\mathrm{K}$ peak to peak, i.e., about 35% of the observed annual cycle in temperature, with water vapour acting to decrease the amplitude of temperature change due to ozone alone, as is to be expected from the relative phases in the individual changes shown in Fig. 3(a) and Fig. 5(a). The cancellation between the ozone and water vapour temperature changes is even stronger

5    at $90\,\mathrm{hPa}$ (Fig. 7(b)) with the combined effect having a peak to peak amplitude of about $2.3 \pm 0.4\,\mathrm{K}$, i.e., again about 35% of the observed annual cycle in temperature. Figure 7(c) gives more information on the vertical structure of the amplitude of the combined temperature effect of ozone and water vapour. Furthermore, Fig. 7(a)–(c) show that whilst the estimated contribution of dynamical heating to the annual cycle in temperatures is substantially smaller than the observed annual cycle, when the contributions from dynamical heating, ozone and water vapour are combined the result is in remarkably good agreement with

10    the observed annual cycle, both in amplitude and in phase. (The maximum and minimum temperature contributions from the dynamical heating are earlier than the observed annual cycle while those from ozone and water vapour are later and the sum has a phase which matches that observed.)

The results in Fig. 7(a)–(c) show that, in combination, ozone and water vapour have a substantial radiative influence on the annual cycle in lower stratospheric temperatures in the TTL and the lower stratosphere above. The estimated radiative effect

15    of ozone and water vapour and the observed annual cycle both peak in amplitude at $70\,\mathrm{hPa}$, but the fractional effect of ozone and water vapour relative to the annual cycle remains substantial below that, down to and including the cold-point, where temperatures control the entry values of stratospheric water vapour. However, as already noted, there are certainly quantitative uncertainties in the estimated radiative effects. The main contributor to this is the uncertainty in the annual mean value of ozone which is about $\pm 0.05\,\mathrm{ppmv}$ at $70\,\mathrm{hPa}$ in the region $20°\,\mathrm{N–S}$. Higher annual mean values of ozone will lead to a smaller





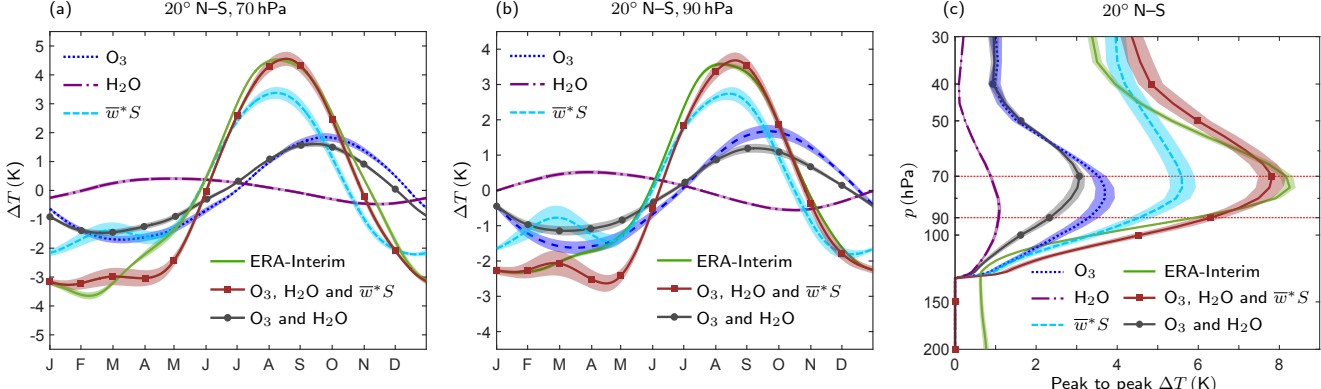

**Figure 7.** Temperature changes (K) calculated using SEFDH method with annual cycles in ozone and water vapour and from the modified SEFDH method (Sect. 4) for the dynamical heating. The plots are averaged between $20°$ N–S at (a) 70 hPa and (b) 90 hPa. The ERA-Interim temperature annual cycle is also shown. Note that the vertical axes are different in (a) and (b). (c) The peak to peak amplitude of the temperature change averaged between $20°$ N–S. Note that there is a phase difference between the temperature from the water vapour and ozone annual cycles. Shadings show 95 % confidence intervals arising from uncertainties in the dataset (see Appendix B for more details).

amplitude in the temperature annual cycle induced by ozone since the opacity of the underlying atmosphere is higher. Further calculations show that this temperature change is not very sensitive to changes in the background value of water vapour and of carbon dioxide within the range of years covered by the SWOOSH dataset.

## 4  Vertical structure of annual cycle

As shown in Fig. 1(c), the temperature annual cycle in the lower stratosphere is largest in a shallow range of pressure levels between about 100 to 50 hPa, with a maximum in amplitude at 70 hPa. This structure has been attributed by Randel et al. (2002) to the long radiative time scales in this region, by appealing to a simplified version of Eq. (1) in which the dynamical heating is specified as a function of height and the temperature-dependent part of the heating is approximated by Newtonian cooling with a radiative time scale that is a function of height. In this section, we revisit the question of what determines the location and structure of the maximum amplitude.

Figure 8(a) shows the variation of $\overline{w}^*$ averaged across the tropics, in height and time, and shows that there is a systematic amplitude decrease with height from 150 to 50 hPa, with no indication of enhanced amplitudes in the 100 to 50 hPa layer. Figure 8(b) shows the full dynamical heating term, $\overline{w}^*\overline{S}$ and Fig. 8(c) shows the same quantity with the annual mean $\langle \overline{w}^*\overline{S} \rangle$ subtracted, to make the annually varying component clear. The amplitude and phase of this quantity has a large contribution from the annual cycle in the upwelling as seen from the plot of the term $(\overline{w}^* - \langle \overline{w}^* \rangle)\overline{S}$ shown in Fig. 8(d). In the region 120 to 90 hPa, the additional contribution from the annual cycle in static stability (the term $\overline{w}^*(\overline{S} - \langle \overline{S} \rangle)$ in Fig. 8(e)), which is roughly out of phase to the upwelling leads to a reduction in the annual cycle in dynamical heating around 100 hPa that comes from the





upwelling component, Fig. 8(d). The resulting annual cycle in dynamical heating, Fig. 8(c), is larger above $100\,\mathrm{hPa}$ than below and rather uniform in amplitude in a deep layer that extends from $90\,\mathrm{hPa}$ up to about $40\,\mathrm{hPa}$, with a relatively broad maximum around $80\,\mathrm{hPa}$. Within this layer the peak to peak amplitude is in the range $0.15$ to $0.2\,\mathrm{K\,day^{-1}}$. In the troposphere the peak to peak amplitude is in the range $0.05$ to $0.1\,\mathrm{K\,day^{-1}}$.

5    To probe these structures further, we now consider a set of SEFDH-like calculations carried out to determine the radiative response to a set of specified dynamical heating structures. These are carried out by perturbing the balance in Eq. (1) by imposing an additional dynamical heating $\Delta\overline{Q}_{\mathrm{dyn}}$ on the right hand side. Note that the peak-to-peak amplitude of dynamical heating shown in Fig. 8(c) does not decrease below the the tropopause as rapidly as does the observed peak-to-peak amplitude of temperature (Fig. 1(a)). To determine whether the localization of temperature variation is a result of the structure of the radiative environment, we first consider an idealized dynamical heating perturbation with no vertical structure, $\Delta\overline{Q}_{\mathrm{dyn}} = -0.1\cos(2\pi\,t/365)\,\mathrm{K\,day^{-1}}$. For this and the next few calculations, we remove the constraint on temperatures below $130\,\mathrm{hPa}$, permitting them to evolve freely in response to the radiative perturbations. Any vertical dependence in the response to this heating will therefore be determined solely by the temperature-dependent part of the radiative heating.

    The resulting temperature change is shown in Fig. 9(a). The amplitude of the response is largest in a layer centred on $100\,\mathrm{hPa}$, roughly where the temperatures are the lowest (Bresser et al. (1995), Fels (1982)). The phase lag with respect to that of the imposed heating is also largest in this layer and equal to about $60\,\mathrm{days}$. This combination of amplitude and phase behaviour would be consistent with a Newtonian cooling model in which the radiative relaxation time scale was a maximum of about $60\,\mathrm{days}$ at $100\,\mathrm{hPa}$, roughly at the cold-point, and decreasing gradually above and below. The implied radiative time scales show a weaker peak than that found by Randel et al. (2002) and in particular do not show such a strong reduction below $100\,\mathrm{hPa}$. One reason for the different results likely arises from the fact that Randel et al. (2002) inferred time scales by looking at the cross-correlation between the annual components of $\overline{T}$ and $\overline{w}^*$ and that includes the extra non-radiative physics operating in the tropical upper troposphere. However following previous authors on this topic, we note that in this region radiative time scales depend strongly on the vertical scale of the perturbation and cannot be precisely defined, therefore there is an inherent imprecision in any explanation of the thermal structure in the TTL based on Newtonian cooling.

    The response to the annual cycle in $\overline{w}^*\overline{S}$ from ERA-Interim (Fig. 8(c)) is now considered. The dynamical heating perturbation is given by $\Delta\overline{Q}_{\mathrm{dyn}} = -(\overline{w}^*\overline{S} - \langle\overline{w}^*\overline{S}\rangle)$ where $\langle\cdot\rangle$ represents the annual mean. We have also assumed, for convenience, that the term $\overline{w}^*\overline{S}$ below $450\,\mathrm{hPa}$ has the same value as at $450\,\mathrm{hPa}$. The precise details do not affect the main conclusions of this calculation. The corresponding temperature response is shown in Fig. 9(b). A comparison of Fig. 9(a) and (b) reveals that the vertical structure in the annual cycle in the ERA-Interim dynamical heating significantly modulates the vertical structure in the temperature response. In particular, the fact that the dynamical heating is larger at $70\,\mathrm{hPa}$ than at $100\,\mathrm{hPa}$ leads to a larger temperature response at $70\,\mathrm{hPa}$ than at $100\,\mathrm{hPa}$, although for given forcing amplitude Fig. 9(a) shows that the response is largest at $100\,\mathrm{hPa}$. Therefore the vertical structure in amplitude of the temperature annual cycle driven by dynamical heating is determined by both the vertical variation of the radiative time scale (to the extent that this quantity can be defined) and the vertical structure of the dynamical heating itself.





For comparison purposes, Fig. 9(c) shows the temperature response to the ERA-Interim dynamical heating if the temperature-dependent part of the radiative heating is represented by Newtonian cooling with a constant radiative relaxation time scale of 40 days. The vertical structure in the annual cycle in $\overline{w}^*\overline{S}$ produces a temperature change that is a good first approximation to that in Fig. 9(b) around 70 hPa, suggesting that the radiative time scale appropriate for the dynamical heating perturbation is

around 40 days, somewhat shorter than that inferred from Fig. 9(a) and consistent with the smaller vertical lengthscale of the imposed perturbation.

Comparing the calculated temperature response to the dynamical heating as shown in Fig. 9(b) to the ERA-Interim temperatures in Fig. 9(d) (same as Fig. 1(a)), there remains below 100 hPa a peak to peak amplitude in the temperature that is significantly larger than is observed. If we assume that this calculation of the dynamical heating provides a reasonable estimate

of the magnitude of the dominant terms in the thermodynamic budget of the upper troposphere, this suggests that upper tropospheric processes provide a stronger constraint on temperature perturbations than do clear-sky radiative processes. Further calculations along the lines of those for Fig. 9(c) suggest (subject to the preceding assumption) that the effective time scale of this constraint is approximately 10 days (not shown).

To illustrate the implications of this tropospheric constraint, we re-introduce the clamp on the temperatures below 130 hPa

so that the annual variation is zero, as a simple representation of these processes. Figure 9(e) shows the resulting temperature change in response to the same dynamical heating perturbation imposed in Fig. 9(b). The reintroduction of the clamp causes the maximum amplitude in temperature to move from around 80 hPa to about 70 hPa and reduces the magnitude of the peak response compared to Fig. 9(b). Whilst we have shown in Sect. 3 that constituent changes below 130 hPa (which in this framework are not allowed to drive local temperature changes) do not have a significant effect on the temperatures at and above

90 hPa, it is clear that temperature changes in this region (or more precisely, the lack of temperature changes in this region) do have a radiative effect on the region above.

We build on this previous calculation and include the perturbations from the ozone and water vapour annual cycles to produce Fig. 9(f). The net effect of the ozone and water vapour annual cycles, as shown in Sect. 3.3, is to increase the annual cycle in temperatures. This produces an annual cycle with a structure that is in better agreement with the ERA-Interim annual cycle,

Fig. 9(d), with a more pronounced peak at 70 hPa. The vertical structure of the amplitude of the temperature response was also shown in Fig. 7(c) (red curve). The inclusion of the annual cycles in ozone, water vapour and dynamical heating produces an annual cycle with an amplitude of $7.8 \pm 0.7$ K at 70 hPa compared to $8.2 \pm 0.3$ K in ERA-Interim (Fig. 7(a)) with good agreement in some months of the year and some differences of up to 1 K between March to May and July to September. Again perturbations to the dynamical heating and constituents are found to be additive.

In summary, the calculations reported in this section suggest that the vertical structure of the peak to peak amplitude in the annual cycle of temperatures arises from a combination of several effects. In the absence of the tropospheric constraints on temperature perturbations, we find that clear-sky radiative processes produce a broad layer centred roughly around 100 hPa over which the relaxation time scales are long. The fact that the amplitude in the temperature response is stronger at 70 hPa than at 100 hPa is a result of the strong tropospheric constraint that appears to be present below 130 hPa, combined with the

vertical structure in the dynamical heating and radiative heating from constituent changes, which both exhibit a peak in the



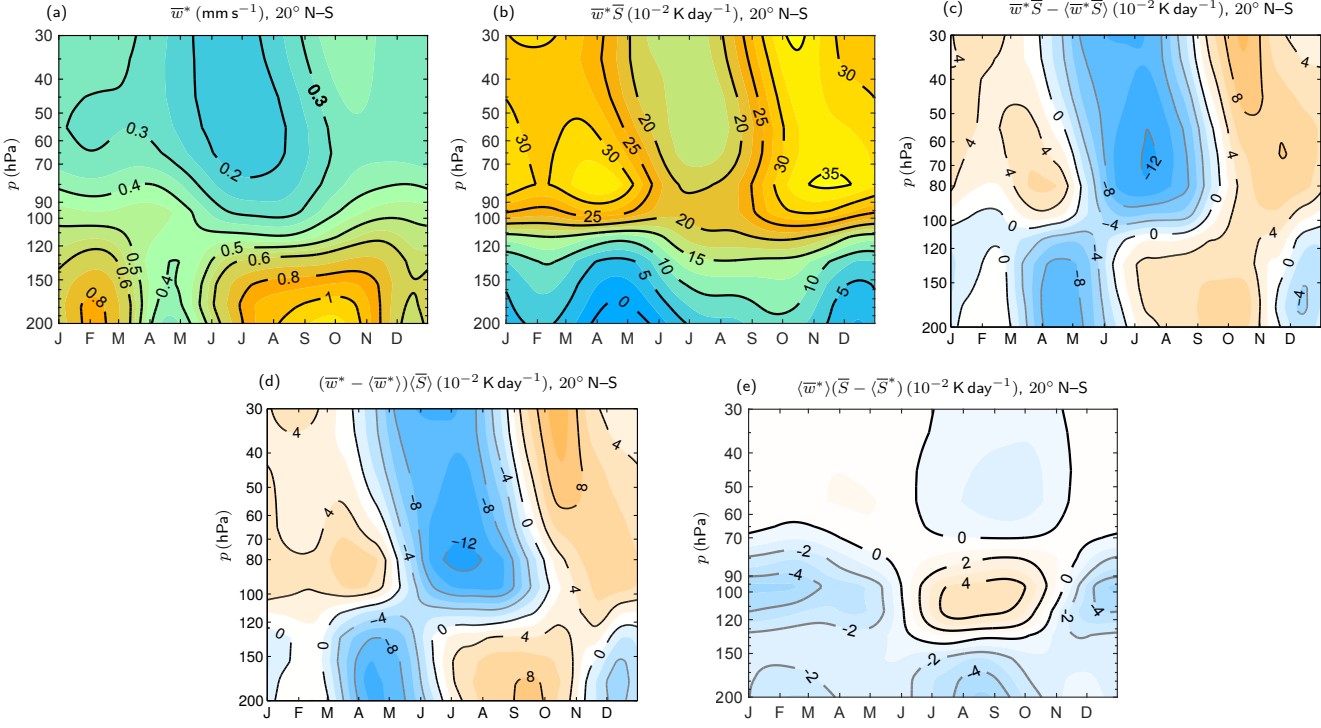

**Figure 8.** Data from ERA-Interim averaged between 1991 to 2010 and 20° N–S. Monthly averages are interpolated to daily values to smooth out the noise in the upwelling field. (a) Mean residual vertical velocity, $\overline{w}^*$. (b) Dynamical heating term $\overline{w}^*\overline{S}$ $(= -\overline{Q}_{dyn})$. (c) Same as (b) but with the annual mean removed. (d) $(\overline{w}^* - \langle \overline{w}^* \rangle)\overline{S}$ component of the dynamical heating. (e) $\langle \overline{w} \rangle (\overline{S}^* - \langle \overline{S} \rangle)$ component of the dynamical heating.

region around 80 to 70 hPa. In our calculations, the combined effects of these processes produce an estimated annual cycle in reasonably good agreement with ERA-Interim.

# 5 Zonally symmetric dynamical adjustment

We will now consider the influence of the annual cycles in ozone and water vapour on the temperatures, allowing for the role of zonally symmetric dynamical adjustments. As noted in Sect. 1, this assumes that there is no change in the wave field and hence in the zonally averaged wave force. The zonally symmetric dynamical adjustment problem has been considered in many previous papers (e.g., Plumb (1982), Garcia (1987), Haynes et al. (1991)). The expectation from this previous work is that the response to an imposed change in constituents, which is equivalent to an applied heating, will occur in part through dynamical heating and the result will be that the temperature response is smaller and smoothed in latitude. The difference from this previous work is that, rather than approximating the temperature-dependent part of the radiative heating by Newtonian




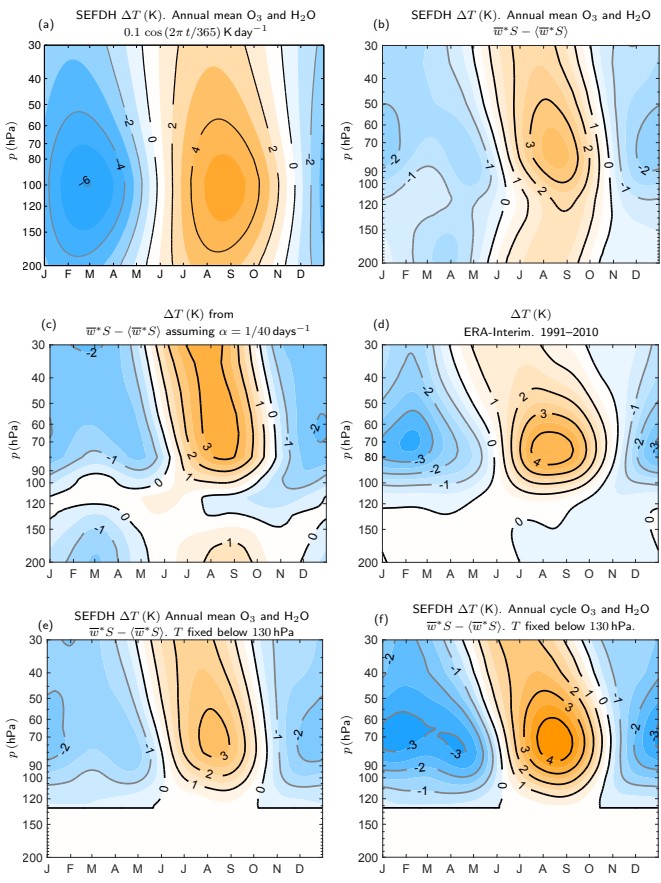

**Figure 9.** Temperature change from the annual mean averaged between $20°$N–S. (a) An SEFDH like calculation with a perturbation of $-0.1\cos(2\pi t/365)\,\mathrm{K\,day^{-1}}$ is added to the dynamical heating with annual mean ozone and water vapour. (b) The temperature change from the annual cycle in the ERA-Interim dynamical heating, $\overline{w}^{*}S - \langle\overline{w}^{*}S\rangle$, shown in Figure 8(c). (c) Temperature change due to the annual cycle in the ERA-Interim dynamical but assuming a constant radiative relaxation rate of $1/40\,\mathrm{days^{-1}}$. (d) ERA-Interim annual mean temperature averaged between 1991 to 2010 (same as Figure 1(a)). (e) Similar to (b) but with temperature held fixed at the annual mean below $130\,\mathrm{hPa}$. (f) Similar to (e) but with the additional perturbation from the annual cycle in ozone and water vapour included.

cooling, we continue to use the modified Morcrette/Zhong and Haigh radiation code. This ensures that our calculations have the maximum possible relevance to those presented in Sect. 3 and 4 and, we would argue, to the real atmosphere.

## 5.1 Model description

For the dynamical calculations, we use the University of Reading IGCM 3.1 [de F. Forster et al. (2000)] which is a hydrostatic primitive equation model based on the original Hoskins and Simmons (1975) spectral dynamical model. Only the coefficients of the zonally symmetric spherical harmonics are retained, up to the total wavenumber 42, resulting in an approximate spacing



of $3°$ between latitudes. There are 60 levels equally spaced in log-pressure coordinates in the vertical with the model top at $50\,\mathrm{km}$. The velocities in the layer near the surface $\sigma > 0.7$ are linearly damped as described in Held and Suarez (1994).

The temperature tendency in the model is given by:

$$\partial_t \overline{T} = [\ldots] + (1 - G(\phi, \sigma))(\overline{Q}_{\mathrm{rad}}(\overline{T}(t), \overline{\chi}(t)) - \overline{Q}_{\mathrm{rad}}(\overline{T}^0, \overline{\chi}^0)) - G(\phi, \sigma)\,\alpha\,(\overline{T} - \overline{T}^0) \qquad (4)$$

where the $[\ldots]$ represents other advective processes in the model and $G(\phi, \sigma) = 0.5(1 + \tanh(50(\sigma - \sigma_{\mathrm{trop}}(\phi))))$. Radiative transfer is imposed above $130\,\mathrm{hPa}$ by setting $\sigma_{\mathrm{trop}}(\phi) = 0.13$. $G(\phi, \sigma)$ is one at the ground and becomes very small towards the top of model. This leads to a smooth transition between the troposphere, which is relaxed to the ERA-Interim annual mean temperature on a time scale $1/\alpha = 10\,\mathrm{days}$, and the stratosphere where the radiative heating rates are determined explicitly by the radiation code. The radiative calculation is implemented in exactly the same way as in the SEFDH calculations in Sect. 3

and 4 except that the shortwave heating is calculated as a diurnal average. We have verified that the standalone radiation code and the version in the model produce consistent longwave and shortwave heating rates.

The control state of the model is set up such that, with the annual mean ozone and water vapour fields from SWOOSH, denoted by $\overline{\chi}^0$ in Eq. (4), the temperature $\overline{T}^0$ remains close to the ERA-Interim annual mean. The additional heating term $\overline{Q}_{\mathrm{rad}}(\overline{T}^0, \overline{\chi}^0)$ in Eq. (4) is needed to maintain these temperatures in the stratosphere. This term is calculated from the radiation

code within the model. The model is initialised in a balanced state with the temperature field equal to the annual mean ERA-Interim values. Although the terms on the right-hand side of Eq. (4) are initially zero, the model state evolves away from the initial state through the effect of dissipative dynamical processes such as surface drag and model hyperdiffusion, to a new steady state. Differences from the initial state are small; for example the temperature difference is less than $2\,\mathrm{K}$ in the tropical stratosphere, and these differences do not affect the results presented below, in particular the comparison of the dynamical

model results to those of the SEFDH calculations.

Perturbation experiments are now carried out by imposing an annual cycle in a trace gas in the same model, with the concentrations read in daily and linearly interpolated to the model time step. Both the control and perturbed runs include a 10-year spin-up period, followed by a further 20 year period. The results shown in the remainder of Sect. 5 are differences between the perturbed and control runs averaged over these 20 years. The nearest model levels to 70, 90 and 100 are 68.8, 87.3

and 98.3 respectively. The figures in this section are shown for these nearest model levels.

### 5.2  Ozone annual cycle

Figures 10(a) and (b) compare the temperature change at $70\,\mathrm{hPa}$ caused by the annual cycle in ozone in the dynamical model and in the SEFDH calculation respectively. (Fig. 10(b) is identical to Fig. 3(b), but is included here for ease of comparison.) The figures show the important effect of including the change in dynamical heating, which tends to broaden the temperature

response in latitude in the tropical region (which is of particular interest in this paper), making it more symmetric about the Equator. Note, in particular, the effect on the off-equatorial maximum at about $10°\,\mathrm{N}$ in the SEFDH calculation, which is no longer a local maximum in the dynamical calculation.



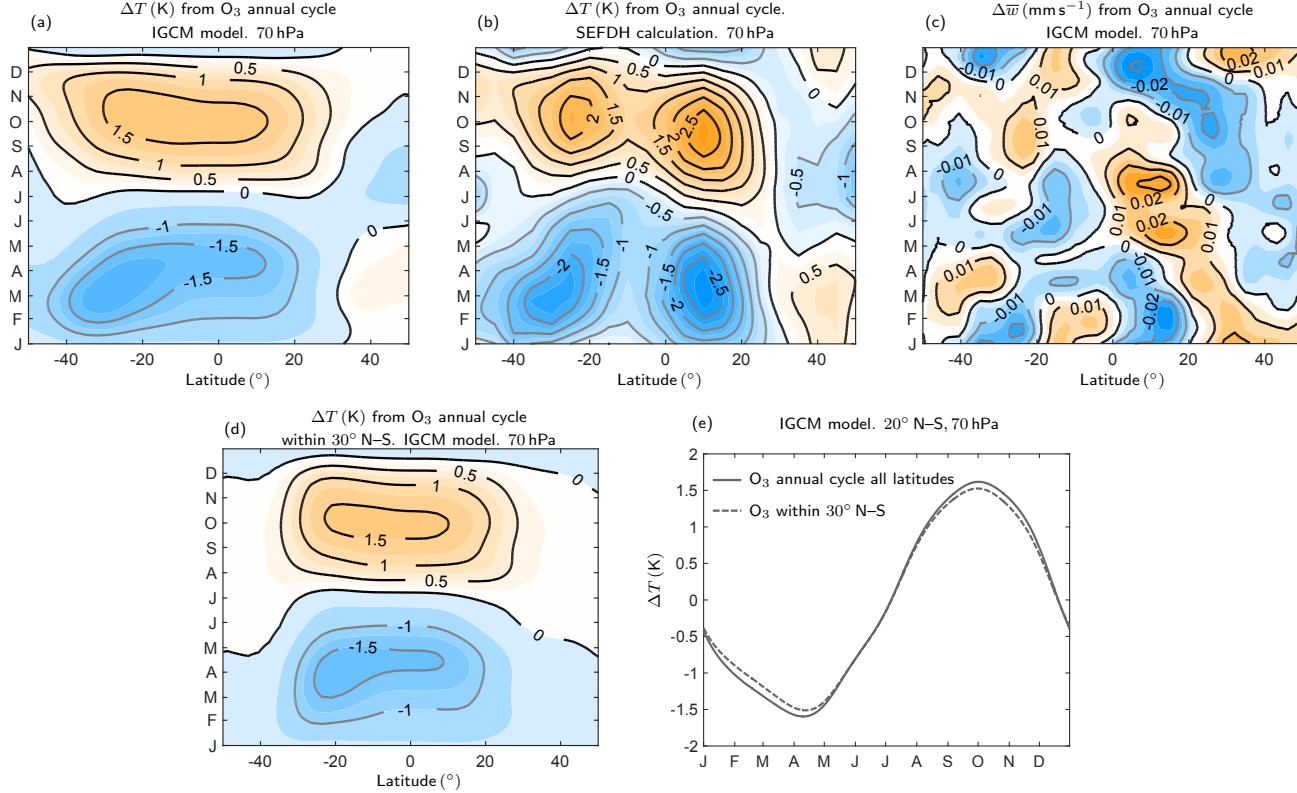

**Figure 10.** (a) Monthly temperature changes showing the annual cycle at 70 hPa calculated using the idealised dynamical model (IGCM) with an annual cycle in ozone. (b) Figure 2(e) is reproduced here for comparison and shows the corresponding SEFDH calculation at 70 hPa. (c) Change in upwelling in idealised dynamical model. (d) Temperature change (K) in an idealised dynamical model calculation with the ozone annual cycle imposed between $30° \text{N–S}$ only (see text for more details). (e) Temperature change at 70 hPa and averaged between $20° \text{N–S}$ calculated with the ozone annual cycle perturbation applied everywhere in the stratosphere (solid line) and only within $30° \text{N–S}$ (dashed line).

This difference between dynamical and SEFDH calculations is as expected from previous theoretical work on the zonally symmetric adjustment problem. The temperature is a dynamical response to a time-dependent applied heating, with the latter equal to the change, with temperatures kept at their control values, in heating due to the change from annual-mean ozone to annually varying ozone. This change in heating field at 70 hPa is shown in Fig. 2(c). Recall the good match between features in Fig. 2(c) and those in the SEFDH-determined temperature change in Fig. 3(b) consistent with the fact that in the SEFDH calculation changes in temperature at each latitude are determined independently (but note also that in neither the SEFDH nor the dynamical calculations is there a local-in-vertical relation between applied heating and temperature response). In the dynamical calculation, on the other hand, the applied heating is balanced in part through a temperature response in both $\partial_t \overline{T}$ and





$\overline{Q}_{\mathrm{rad}}$, and in part by a response in dynamical heating (principally $\overline{w}^* \overline{S}$). Both responses are determined by coupling between latitudes and it is this that is the crucial difference from the SEFDH framework.

The $\overline{w}^*$ response at $70\,\mathrm{hPa}$ to the ozone variation is shown in Fig. 10(c). Note that the $\overline{w}^*$ field tends to emphasize the smaller latitudinal scale features in the heating field shown in Fig. 2(c), e.g., the two regions of strong negative heating at about $30°\,\mathrm{S}$

and about $10°\,\mathrm{N}$ in January and February and the regions of strong positive heating at about $20°\,\mathrm{S}$ in September and October and at about $10°\,\mathrm{N}$ in August and September. On the other hand between these regions of strong positive or negative heating there tends to be an opposite signed feature in $\overline{w}^*$. The overall effect is that the corresponding dynamical heating, which is opposite in sign to $\overline{w}^*$, tends to broaden the temperature response relative to the pattern of heating.

The non-locality in latitude in the dynamical problem means that the temperature response in the tropics, shown in Fig. 10(a),

is potentially determined in part by the change in ozone in the extratropics. To quantify this effect, we restrict the ozone annual cycle perturbation to the tropical region between $30°\,\mathrm{N–S}$ using a mask of the form $0.5\left[\tanh((\phi+\phi_1)/c)-\tanh((\phi-\phi_1)/c)\right]$ where $\phi_1 = 30°$ and $c = 2°$. The results are shown in Fig. 10(d) and (e). The net effect of the annual variation in ozone in the extratropics is to increase the amplitude of the temperature response in the tropics from $3.0\,\mathrm{K}$ to $3.2\,\mathrm{K}$ peak to peak. The dominant contribution from ozone to the annual cycle change in temperature in the tropics is therefore from ozone in the tropics.

The contribution from ozone in the extratropics arises because of the effects of time dependence (e.g, Garcia (1987), Haynes et al. (1991)), with shorter time scales relative to radiative time scales implying coupling over larger latitudinal distances. The time scale of the annual cycle variation is sufficiently long (relative to radiative time scales) that the effect of extratropical ozone on the tropical temperatures is weak.

### 5.3   Water vapour annual cycle

The temperature response of the dynamical model to a perturbation from annual average water vapour to annually varying water vapour is now considered in a similar way to the ozone perturbation just discussed. Given the significant radiative interactions in the water vapour response between different vertical layers, the temperature responses at each of the levels $70\,\mathrm{hPa}$, $90\,\mathrm{hPa}$ and $100\,\mathrm{hPa}$ are displayed respectively in Fig. 11(a)–(c). The corresponding SEFDH temperature responses at $70\,\mathrm{hPa}$, $90\,\mathrm{hPa}$ and $100\,\mathrm{hPa}$ are shown respectively in Fig. 11(d)–(f). As was the case for ozone, the temperature response in the dynamical model,

shown in Fig. 11(a)–(c), is a broadening of the SEFDH temperature response. The prominent maxima in positive heating in March and April at $20°\,\mathrm{N}$ at $70\,\mathrm{hPa}$ and about $25°\,\mathrm{N}$ at $90\,\mathrm{hPa}$ and $100\,\mathrm{hPa}$, and in negative heating in September to November at the same locations, are reduced in magnitude, but over the Equator and extending into the SH there is increased positive heating in March and April and increased negative heating in September to November. The resulting structure is therefore, in the tropics, much more symmetric across the Equator than the SEFDH temperature response.

As for ozone, there is the possibility that extratropical changes in water vapour have an effect on tropical temperatures, and this was investigated as before, by applying a latitudinal mask to the perturbation to water vapour. Also as for ozone, with the same explanation, the effect of extratropical water vapour variations on tropical temperatures were found to be weak, reducing the peak to peak temperature variation averaged over $20°\,\mathrm{N–S}$ from $1.1\,\mathrm{K}$ to $1.0\,\mathrm{K}$ (not shown).





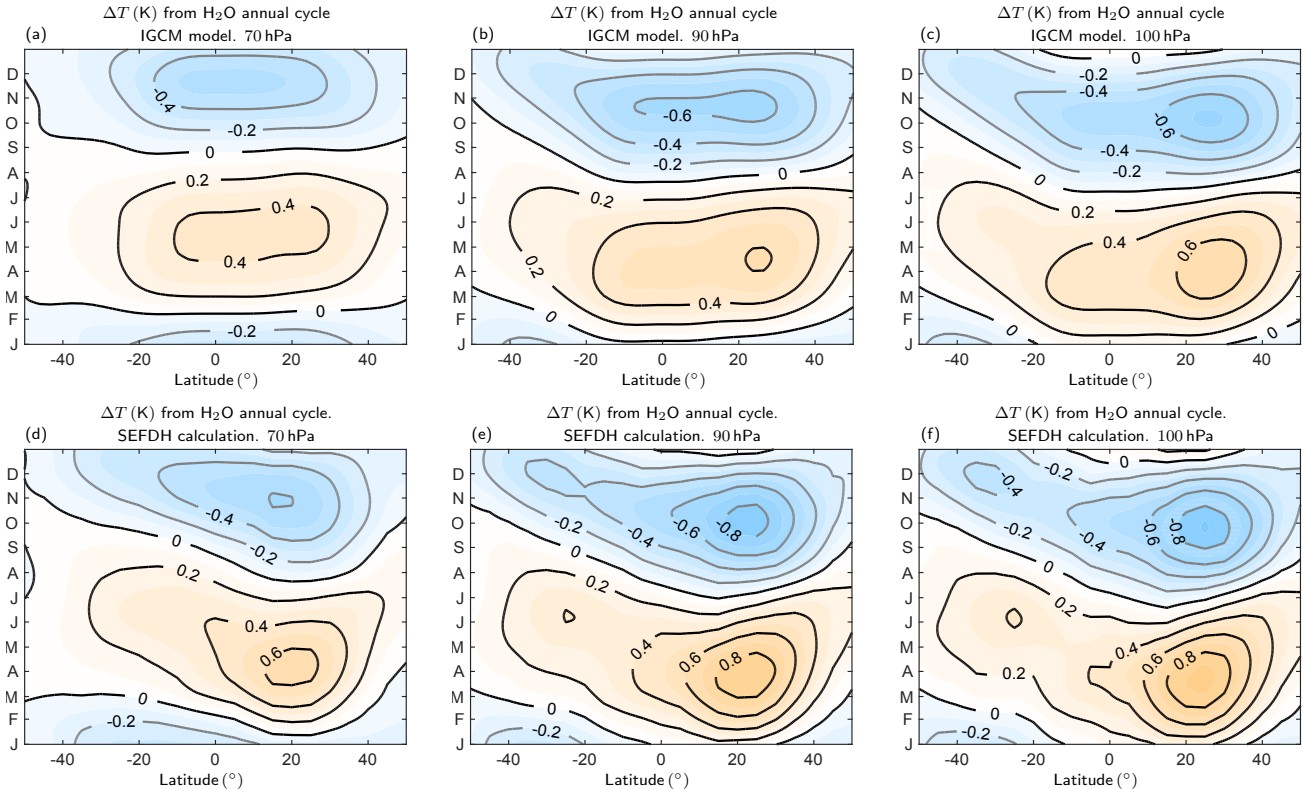

**Figure 11.** Temperature changes (K) calculated using the idealised dynamical model (IGCM) with an annual cycle in water vapour at (a) 70 hPa, (b) 90 hPa and (c) 100 hPa. Temperature change from the SEFDH calculation (same as Fig. 5(b)–(d)) at (d) 70 hPa, (e) 90 hPa and (f) 100 hPa for comparison.

## 5.4 Ozone and water vapour annual cycles

The combined effect of the ozone and water vapour annual cycles in the dynamical calculation is now considered. The individual temperature responses, to very good approximation, add linearly. The latitudinal structure of the combined response, which may also be deduced, for example, by combining Fig 10(a) and 11(a) is shown in Fig. 12(a) for 70 hPa. The response
5  at 90 hPa is also shown in Fig. 12(b). Fig. 12(c) and (d) show the temperature responses in the dynamical model averaged between 20° N–S, to ozone and water vapour individually and the combined response, at 70 hPa and at 90 hPa respectively. Also shown on these figures are the SEFDH results (same as Fig. 7(a) and (b)). By this tropical average measure, there is negligible reduction in the peak to peak amplitudes of the individual and combined temperature responses to ozone and water vapour variations, in going from the SEFDH calculation to the dynamical calculation. Any reduction in local latitudinal maxima in
10  the temperatures response is offset by the broadening effect, leaving the tropical average essentially the same. However, we reiterate that important changes in the structure of the temperature response across the tropics occur as a result of including the zonally symmetric dynamical adjustment.


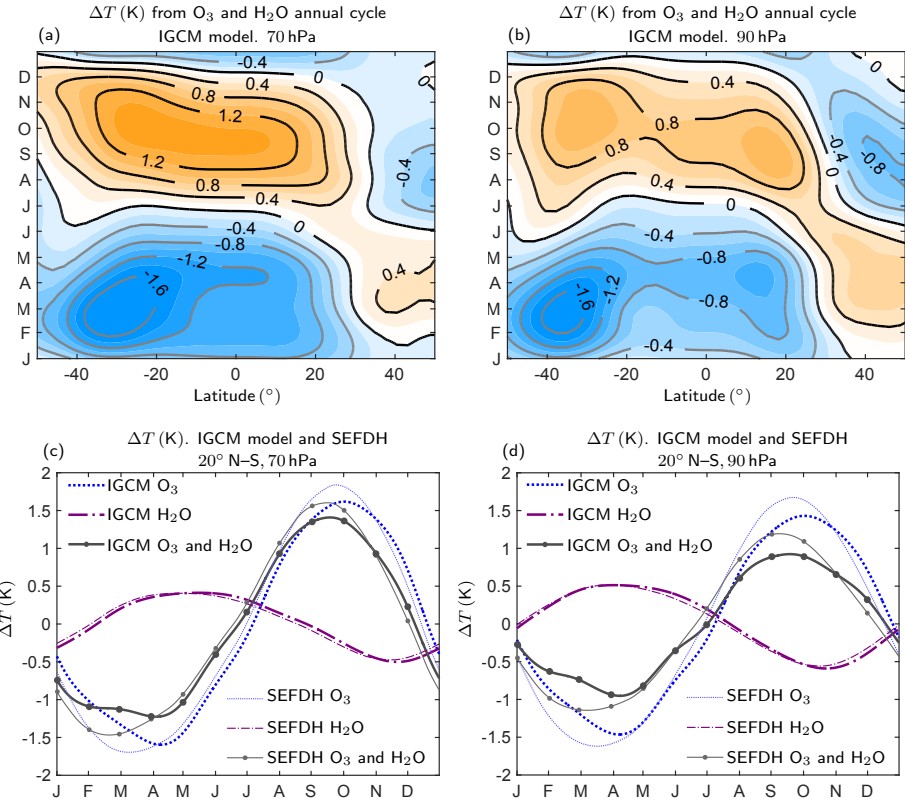

**Figure 12.** Temperature changes (K) calculated using the idealised dynamical model (IGCM) with annual cycles in both ozone and water vapour shown at (a) $70\,hPa$ and at (b) $90\,hPa$. Temperature changes averaged between $20°\,N–S$ at (c) $70\,hPa$ and (d) $90\,hPa$ and showing the effects of ozone and water vapour in the dynamical model (thick lines) as well as the corresponding SEFDH temperature changes from fig. 7(a) and (b) (thin lines).

## 6 Discussion

In this work, we have analysed radiative aspects of the prominent annual cycle in temperature in the TTL/tropical lower stratosphere. The cycle has the largest peak to peak amplitude at $70\,hPa$ ($\sim 8\,K$) but is also significant at lower levels (e.g., peak to peak amplitudes of $\sim 6\,K$ at $90\,hPa$ and $\sim 3\,K$ at $100\,hPa$ respectively) where it has a direct effect on concentrations

5   of water vapour in air entering the stratosphere. The amplitudes quoted here are on the basis of the ERA-Interim dataset, with temperatures averaged from $20°\,N$-S, but these are in good agreement with other current and earlier datasets.

Building on previous work that suggested an important contribution to the annual cycle in temperatures from the radiative effects of the annual cycle in ozone (Folkins et al. (2006), Randel et al. (2007), Chae and Sherwood (2007), Fueglistaler et al. (2011)), we have calculated the temperature response to the annual variation in ozone and water vapour, which are

10   both important radiatively active trace gases in this part of the atmosphere. During the review process, we were also made



aware of the work of Gilford and Solomon (2016, submitted) concerning the ozone contribution to the temperatures using an SEFDH calculation. We extend this previous work by presenting explicit results for the effect of water vapour variation and by paying particular attention to the vertical structure of the temperature response and the role of variations in the trace gas concentrations in different vertical layers. In our first approach, we have used a seasonally evolving fixed dynamical heating

(SEFDH) calculation in which the temperature response to annual variations in a trace gas is calculated independently at each latitude, assuming that the dynamical heating at each height is unchanged from its value in a control state in which the trace gas concentrations are constant (and equal to their annual mean values). We have taken concentrations of ozone and water vapour as zonal averages from the recently released SWOOSH dataset (Davis et al. (2016)).

We find substantial contributions to the annual cycle in temperatures from the annual variations in ozone: $3.5 \pm 0.4\,\mathrm{K}$ at

$70\,\mathrm{hPa}$, $3.3 \pm 0.5\,\mathrm{K}$ at $90\,\mathrm{hPa}$ and $2.6 \pm 0.2\,\mathrm{K}$ at $100\,\mathrm{hPa}$ (all values quoted from here on are peak to peak amplitudes averaged between $20°\,\mathrm{N\text{-}S}$) and from water vapour: $0.9 \pm 0.1\,\mathrm{K}$ at $70\,\mathrm{hPa}$, $1.1 \pm 0.1\,\mathrm{K}$ at $90\,\mathrm{hPa}$ and $0.99 \pm 0.03\,\mathrm{K}$ at $100\,\mathrm{hPa}$.

Whilst the ozone contribution maximises around $70\,\mathrm{hPa}$ and is roughly in phase with the observed temperature annual cycle, the water vapour contribution maximises around the cold-point at $90\,\mathrm{hPa}$ and is of the opposite phase. We therefore conclude that the radiative effects of ozone variations are increasing the amplitude of the temperature annual cycle (over the value it

would have in the absence of such variations) and, in the same sense, the radiative effects of water vapour variations are reducing it. Despite the cancellation, the net effect of variations in ozone and water vapour together is substantial and amounts to about 35% of the observed annual cycle at both $70\,\mathrm{hPa}$ and $90\,\mathrm{hPa}$ and about 45% at $100\,\mathrm{hPa}$ (Fig. 7).

Further SEFDH calculations showed that the ozone-induced temperature variation at $70\,\mathrm{hPa}$ is caused primarily (80%) by ozone variation in the region 90 to $50\,\mathrm{hPa}$ (Fig. 3), i.e., ozone variations local to $70\,\mathrm{hPa}$. In contrast, the water vapour induced

temperature variation is caused by both local and non-local water vapour variation, e.g., 60% of the water vapour induced temperature variation at $90\,\mathrm{hPa}$ comes from water vapour variation in the region 100 to $80\,\mathrm{hPa}$ and 40% from the region 130 to $100\,\mathrm{hPa}$. This upward non-local radiative effect is seen throughout the lower stratospheric region and has important implications for cold-point temperatures. For example, in the current configuration, where the annual cycle in water vapour at the cold-point and below the cold-point have similar phase, if the amplitude of the annual cycle in water vapour below the cold-point was to

increase, then the radiative effect would be to reduce the amplitude in the annual cycle in cold-point temperatures and hence reduce the amplitude of the annual cycle in water vapour, i.e., the amplitude of the 'tape recorder' signal, at and above the cold-point.

We also examined the factors controlling the vertical structure of the annual cycle in temperatures. The observed structure, a maximum centred on $70\,\mathrm{hPa}$ and largely restricted to the 50 to $100\,\mathrm{hPa}$ layer, arises from a combination of several factors. The

vertical structure in the annual cycle in dynamical heating, which maximises around $80\,\mathrm{hPa}$, is important (Fig. 8(c)). In a simple radiative calculation where the temperatures are allowed to vary at all heights, this structure in the dynamical heating alone leads to a maximum in the temperature annual cycle at around $70\,\mathrm{hPa}$ with a broader vertical structure than observed, Fig. 9(b). In particular, the calculated temperature annual cycle is much larger than observed in the region below $130\,\mathrm{hPa}$, the reason being that non-radiative upper tropospheric processes operate to limit the temperature variations. Clamping the temperatures below

$130\,\mathrm{hPa}$ in the radiative calculation leads to a temperature response to dynamical heating (Fig. 9(e)) that decreases downward



more rapidly from the maximum at 70 hPa. Additionally the temperature annual cycle caused by the combined ozone and water vapour variations has maximum amplitude around 70 hPa (Fig. 7(c)), reinforcing the effect of the dynamical heating.

In the last part of this paper, we investigate the effect of relaxing the SEFDH assumption, thereby going beyond the work of Fueglistaler et al. (2011), for both the ozone and water vapour variations. We do this by incorporating the radiative code used for the SEFDH calculations within a 2D (height-latitude) dynamical model, i.e., we take account of the details of radiative transfer without any simplifying assumption such as Newtonian cooling. We find that for both the ozone and water vapour, part of the heating associated with annual variation in concentrations drives an annual cycle in the upwelling. A change in $\overline{w}^*$ is induced within the 2D zonally symmetric dynamics by the change in heating and tends to have small latitudinal scales (Fig. 10(c)). Although the magnitude of the change in upwelling is small (Fig. 10(c)) compared to that in ERA-Interim (Fig. 8(a)), the corresponding dynamical heating is important in cancelling the small latitudinal scales in the change in radiative heating from the change in constituents, therefore smoothing the temperature response in latitude. The overall effect on the temperature response averaged over 20° N–S, where the annual cycle is largest, is rather small, for each of ozone and water vapour. The non-locality in latitude of the dynamical response means that in principle the 20° N–S region can be affected by the radiative effect of ozone and water vapour variations outside of this region. However detailed examination showed that this non-local (in latitude) effect is very small. Therefore the overall conclusions from the SEFDH calculation on contributions, through radiative effects, of tropical ozone and water variations to tropical temperature variations averaged over the 20° N–S tropical region are robust to including the dynamical adjustment, in particular the conclusion that the net effect of ozone and water vapour contribute to about 35% of the annual cycle peak to peak value at 70 hPa and 90 hPa respectively. In contrast, the detailed latitudinal structure predicted by the SEFDH calculation is not robust.

Within the 2D zonally symmetric dynamical formalism presented here we do not take account of changes in wave-induced forces and consequent changes in $\overline{w}^*$ which would be needed for a full dynamical calculation of the radiative effects of ozone and water vapour variations. This effect has been discussed by several authors over the last 30 years or so, including Fels et al. (1980) and Garcia (1987), usually making the assumption that the wave force can be represented by Rayleigh friction (so that local wave force is proportional and opposite to the local zonal velocity). However, it is generally accepted that Rayleigh friction is a poor representation of the wave forces that operate in the upper troposphere and stratosphere. Ming et al. (2016) analyse the effect of the change in wave force in the response to imposed steady localised zonally symmetric heating in a simple 3D model (where the waves are resolved and no Rayleigh friction assumption is necessary) and argue that the effect is to broaden the temperature response, particularly at low latitudes. Latitudinal structure in the imposed heating tends to be balanced by the dynamical heating associated with the meridional velocity response and the change in wave force provides the necessary angular momentum balance. A similar effect is seen in the zonally symmetric problem with Rayleigh friction (Garcia (1987), e.g., see Fig. 6 in their paper). There is an analogous effect in the time-dependent zonally symmetric response problem, without any change in wave force or Rayleigh friction, considered in Sect. 5, with the angular momentum balance including the zonal acceleration. Therefore the effect of including the change in wave force in the dynamical problem is, broadly speaking, expected to be similar and in addition to that already seen in the time-dependent zonally symmetric problem, that at low latitudes the dynamical response will smooth the temperature response to latitudinally varying heating (caused by



latitudinal variations in radiatively active constituents). If the change in wave force is weak (or in the corresponding Rayleigh friction problem the friction coefficient is much less than the annual frequency) then the additional effect will be small. If the change in wave force is strong then the result will be that the smoothing is over a larger range of latitudes. The fact that the observed annual cycle in temperature is coherent over the latitude range 20° N–S, but no more than that (recall Fig. 1(b)–(d))

suggests that the wave force effect cannot be too strong. Therefore we expect that our conclusions from the zonally symmetric dynamical problem studied here would not be changed too much if the change in wave force was included.

Notwithstanding the above uncertainty, one reason for the purely 2D zonally symmetric calculation being valuable is in the diagnostic interpretation of reanalysis and model data. From such data it is possible to calculate the seasonal variation in wave forces (i.e., the divergence of the Eliassen-Palm flux) and to consider the meridional circulation (e.g., Randel et al. (2008)) and

temperature response to those forces. The difference between the actual seasonal variation of temperature (in the reanalysis data or in the model) and the calculated response might be potentially be explained by the meridional circulation or temperature response to the variations in heating caused by seasonal variations in ozone and water vapour. Since the effect of ozone and water vapour heating on the wave forces is already included in the wave forces extracted from the data, the relevant dynamical problem to predict these responses is the 2D problem.

More generally, the differences between the temperature responses to ozone and water vapour calculated through the SEFDH approach and those calculated using the 2D dynamical model (and for reasons given above this conclusion is likely to extend to a more general dynamical response) demonstrate that low-latitude temperature features with small latitudinal scales predicted by SEFDH calculations should not be taken too seriously, because these features will be smoothed out by the dynamical response. This applies to Fig. 3(b) and 5(b)–(d) in this paper, to previous SEFDH calculations of the temperature response

to annual variations in ozone (Fueglistaler et al. (2011), their Fig. 5(b)) and to similar calculations of the effect of recent interannual variations in ozone and water vapour (Gilford et al. (2016), their Fig. 6).

Current comprehensive global (chemistry-)climate models show a large spread in the amplitude of the TTL annual cycle in temperature (e.g., Kim et al. (2013)), but the quantitative causes of these differences are not well understood. The results of this study show that an erroneous representation of the annual cycles in ozone and water vapour, as is commonplace amongst

such models (e.g., Gettelman et al. (2010)), is likely to be a major contributor to poor model performance for capturing the TTL temperature annual cycle. Similar conclusions are likely to apply to interannual variations, e.g., in the 2010–2013 period investigated by Gilford et al. (2016) using SEFDH calculations. Progress in improving the representation of the TTL in comprehensive global models therefore requires consideration of the coupling through transport and radiative effects between dynamics, ozone and water vapour in the TTL. Specific aspects highlighted by our results include a strong sensitivity of ozone

radiative effects to mean ozone concentrations in the 90 to 70 hPa region, for which models with interactive chemistry simulate a range of values (Gettelman et al. (2010)) and for which a range of observation-based gridded datasets exist for climate models that do not include chemistry (Cionni et al. (2011), Bodeker et al. (2013)). Furthermore, because of the importance shown here of non-local radiative effects for water vapour in the TTL, modelled cold-point temperatures are likely to be sensitive to the representation of water vapour concentrations in the upper tropical troposphere.





## Appendix A: FDH calculations

A first order estimate of the effect of specified perturbations to radiative trace gases on temperatures in the TTL and the stratosphere can be made using a fixed dynamical heating (FDH) calculation where it is assumed that the dynamical heating remains constant from the unperturbed to the perturbed state, i.e., that no changes in circulation occur as a result of the perturbation (Fels et al. (1980), Ramanathan and Dickinson (1979)). The time scale for stratospheric adjustment to the perturbation is essentially the stratospheric radiative damping time. This is about $40\,\mathrm{days}$ in the tropical lower stratosphere and less than a week near the stratopause, although different techniques estimate different values and furthermore the time scale is dependent on the vertical scale of the heating perturbation (e.g., Dickinson (1973), Mlynczak et al. (1999), Hitchcock et al. (2010)). These stratospheric time scales are relatively short compared to that required for tropospheric temperatures to adjust to the perturbation, because these are strongly constrained to surface temperatures, which particularly in oceanic regions, will evolve only on time scales of months or years. Hence, in FDH calculations, temperatures are held fixed below some level, often corresponding to the (radiative) tropopause. We choose this level to be $130\,\mathrm{hPa}$, consistent with previous calculations. The reasons for this choice are justified in Sect. 3.

The FDH calculation is a simplified version of the SEFDH calculations and the equations below can be compared to Eq. (2) and (3). Given the background profiles of temperatures and concentrations of trace gases ($\overline{T}_0, \overline{\chi}^0_{O_3}, \overline{\chi}^0_{H_2O}$), the dynamical heating, $\overline{Q}^0_{dyn}$, is first calculated by assuming the balance

$$\overline{Q}_{rad}(\overline{T}^0, \overline{\chi}^0_{O_3}, \overline{\chi}^0_{H_2O}) + \overline{Q}^0_{dyn} = 0. \tag{A1}$$

The dynamical heating is not a function of time unlike the SEFDH calculations. A perturbation is then applied to trace gas concentrations ($\Delta\overline{\chi}_{O_3}, \Delta\overline{\chi}_{H_2O}$) and the equilibrium temperature state, $\overline{T}^0 + \Delta\overline{T}$, is obtained from

$$\overline{Q}_{rad}(\overline{T}^0 + \Delta\overline{T}, \overline{\chi}^0_{O_3} + \Delta\overline{\chi}_{O_3}, \overline{\chi}^0_{H_2O} + \Delta\overline{\chi}_{H_2O}) + \overline{Q}^0_{dyn} = 0. \tag{A2}$$

Time averaged profiles of ozone and water vapour from the SWOOSH dataset and the annual mean temperature from ERA-Interim at the Equator are used as the base profile and the trace gases are then perturbed. The calculation is done at the Equator on January 1 and the albedo is set to 0.085 (these details are relevant for the shortwave heating). The 100 pressure levels used in all radiative calculations are: 1, 2, 3, 4, 5, 6, 7, 8, 9, 10, 11, 12, 13, 14, 15, 16, 17, 18, 19, 20, 22, 25, 27, 30, 35, 40, 45, 50, 55, 60, 65, 70, 75, 80, 85, 90, 93, 95, 97, 100, 103, 105, 107, 110, 113, 115, 117, 120, 123, 125, 127, 130, 133, 135, 137, 140, 145, 150, 155, 160, 165, 170, 175, 180, 185, 190, 200, 205, 210, 215, 220, 225, 230, 235, 240, 245, 250, 255, 260, 265, 270, 275, 280, 285, 290, 295, 300, 320, 330, 340, 350, 370, 400, 450, 500, 600, 700, 800, 900, 1000 hPa.

Numerically, the FDH calculation is done by iterating the temperatures forward with a time step of one day using the longwave heating rates to find the new equilibrium temperature. The values are considered to have converged when the temperature change and the fluxes between pressure levels after consecutive time steps falls below $5 \times 10^{-4}\,\mathrm{K}$ and $1 \times 10^{-7}\,\mathrm{K\,m^{-1}}$ respectively. In practice, these thresholds are reached after about 500 days which is much larger than any radiative time scales in the stratosphere, hence, ensuring that the temperatures in the stratosphere have converged.



In Sect. A1 and A2 below, we describe in detail the temperature response to example perturbations in ozone and in water vapour. These provide helpful background for understanding the response to the annual cycle in these two gases reported in Sect. 3.

**A1  Ozone perturbation**

5   The example perturbation applied to ozone concentrations is a reduction in concentrations in the lower stratosphere (solid line in Fig. 13(a) left). This is a simple representation of lower stratospheric concentrations in NH winter, relative to the annual mean. The perturbation is a Gaussian of the form $A_0 \exp[-0.5((z-18.6)/2)^2]$ where $A_0 = -0.07$ (ppmv) and $z = -7\log(p/1\times10^5)$ km. Removing ozone in the lower stratosphere leads to an instantaneous local decrease in the longwave and shortwave heating (Fig. 13(a) right) and results in a local decrease in the temperature in an FDH calculation, Fig. 13(b)
10  (where 'local' refers to the vertical region in which the perturbation in concentrations is applied).

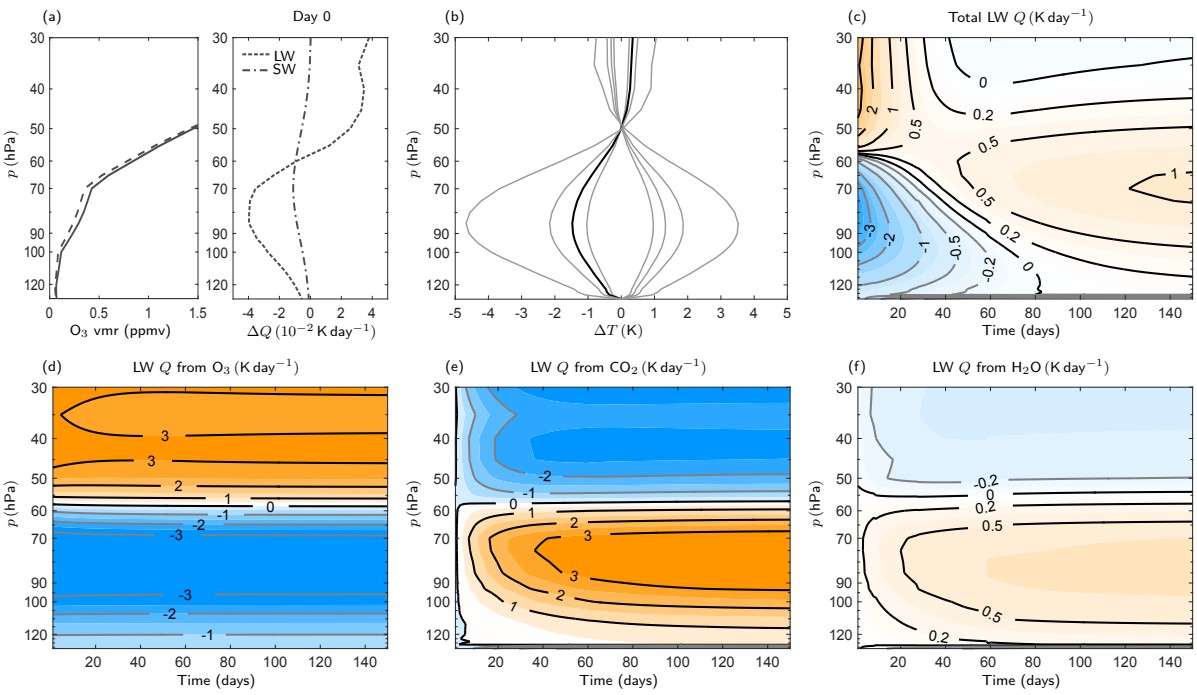

**Figure 13.** (a) Left: ozone reference profile (solid) and perturbed profile (dashed) used in the FDH calculation. Right: instantaneous change in heating rate from perturbation. (b) Temperature change resulting from the ozone perturbation. The calculation is done at the Equator on January 1. The lines correspond to perturbations of $A_0 =$-0.2, -0.1, **-0.07** (black line), -0.05, 0.05, 0.07, 0.1 and 0.2 ppmv from left to right at the maximum temperature change (see text for more details). For the perturbation of $-0.7$ ppmv, (c) total longwave heating rate from all the constituents, (d) longwave heating rate due to ozone, (e) longwave heating rate due to carbon dioxide and (f) longwave heating rate due to water vapour.





The time evolution of various components of the longwave radiative heating after the perturbation is applied are shown in Fig. 13(b)–(e), respectively total longwave heating and then the individual contributions from ozone, carbon dioxide and water vapour. The instantaneous effect of the reduction in ozone concentrations is to cause a decrease in both the shortwave heating, and the local longwave heating, because of reduction in local longwave absorption. The shortwave change, which

has peak amplitude $-1.1 \times 10^{-2}\,\mathrm{K\,day^{-1}}$, occurs because of reduced shortwave absorption and is essentially proportional to the local change in concentration. The instantaneous longwave change is significantly larger, with peak amplitude $-4 \times 10^{-2}\,\mathrm{K\,day^{-1}}$, and, in addition to the local decrease, there is an increase, with similar peak amplitude in the region above the concentration perturbation (see Fig. 13(d)). The explanation for this vertical structure is that, because ozone concentrations are small in the troposphere, in the lower stratosphere there is a substantial upwelling flux of longwave radiation of wavelength

relevant to ozone ($9.6\,\mu$m-band), and the imposed perturbation in ozone concentrations leads to less local absorption of this upwelling radiation with, correspondingly increased absorption above the perturbation. Note that another potential effect of the perturbation to ozone concentrations is reduced local emission which would imply a local heating. Figure 13(d) shows that any effect of change in emission is dominated by the changed absorption of upwelling radiation.

In the response to the instantaneous change in heating just described, the temperature and hence the longwave fluxes change,

with both carbon dioxide (Fig. 13(e)) and to a lesser extent water vapour (Fig. 13(f)) contributing significantly. Note that changes in the ozone longwave heating, after the instantaneous change resulting from the perturbation to ozone concentrations, are weak, suggesting that it plays little role in the temperature adjustment. An equilibrium is reached where the net longwave heating (Fig. A1(c)) balances the reduction in shortwave heating. The equilibrium temperature change is dominated by a local decrease centred on $70\,\mathrm{hPa}$ (i.e., the centre of the region where ozone concentrations were perturbed). Several time scales are

involved in the adjustment process and Fig. 13(c) shows that the heating rates and hence the temperature are still evolving after 100 days. This justifies the use of an SEFDH rather than an FDH calculation when studying the annual cycle in temperatures.

Further experiments show that the FDH temperature response varies approximately linearly with the peak value of the Gaussian perturbation in the range $-0.1$ to $0.1\,\mathrm{ppmv}$ (thin grey lines in Fig. 13(b)), so that the detailed time evolution described above continues to hold if heating and temperature anomalies are multiplied by the appropriate factor. In particular a modest

increase in ozone concentrations will lead to a local temperature increase, in which the net (negative) change in longwave heating balances an increase in shortwave heating. For concentration anomalies with peak values of $\pm 0.2\,\mathrm{pmmv}$ significant non-linear effects appear.

## A2  Water vapour perturbation

Following the approach in Appendix A1 above, a corresponding calculation is now described in which water vapour is per-

turbed by removing a Gaussian of the form $B_0 \exp[-0.5((z-16.9)/1.5)^2)]$ where $B_0 = 1.0\,(\mathrm{ppmv})$ (Fig. 14(a) left) which leads to an instantaneous local decrease in the shortwave and a local increase in the longwave radiation (Fig. 14(a) right). This is also a very simple representation of lower stratospheric concentrations in NH winter, relative to the annual mean. As in Appendix A1, Fig. 14(c)–(f) respectively show the total longwave heating and then the individual contributions from ozone, carbon dioxide and water vapour, during the evolution in response to the water vapour perturbation.





The abundance of water vapour in the troposphere means it is relatively opaque to upwelling longwave radiation in the main water vapour absorption bands. This means that, in contrast to ozone, the dominant instantaneous effect in the longwave of locally reducing the water vapour in the lower stratosphere, is to cause less local emission, i.e. local heating, and, corresponding, less non-local absorption in neighbouring regions, i.e., non-local cooling, rather than any effect on absorption of upwelling radiation. This can be seen in the water vapour longwave heating shown in Fig. 14(f). Note that the change in non-local absorption is seen primarily in the upper troposphere below the region where the concentrations are reduced, presumably because background water vapour concentrations are relatively large there compared to those in the stratosphere. The reduction in water vapour concentration also leads to a reduction in shortwave absorption, as was the case for ozone, but the magnitude $(-0.3 \times 10^{-2}\,\mathrm{K\,day^{-1}})$ is smaller than the corresponding change in longwave heating $(4.3 \times 10^{-2}\,\mathrm{K\,day^{-1}})$.

In the evolution following the initial instantaneous change in heating the longwave heating contributions to due to carbon dioxide, water vapour and ozone all play a role to limit the temperature response and redistribute it in the vertical. (Fig. 14(d)–(f)). In particular the initial local increase in temperatures is transmitted in the vertical through longwave fluxes in the carbon dioxide bands to give subsequent temperature increases significantly above the layer in which water vapour concentrations were perturbed. This sort of behaviour could not be captured by a local Newtonian cooling approximation. As was the case for ozone, the longwave heating (and hence the temperatures) continue to evolve beyond $100\,\mathrm{days}$. This suggests that a sequence of quasi-steady FDH calculations would be inadequate for studying the annual cycle in temperatures and justifies the use of the SEFDH approach.

Experiments with different amplitudes of perturbation to water vapour concentration (Figure 19(b)) show that the response is linear for peak values up to $\pm 1.0\,\mathrm{ppmv}$, with non-linear effects visible at $\pm 2.0\,\mathrm{ppmv}$. Note that a similar amplitude and shape of perturbation as the ozone perturbation with $A_0 = 0.2\,\mathrm{ppmv}$ is shown for comparison as a dashed grey line in Fig. 14(b) and the magnitude of the temperature change is small ($0.14\,\mathrm{K}$ at $70\,\mathrm{K}$) compared to that for the equivalent ozone perturbation ($2.8\,\mathrm{K}$ at $70\,\mathrm{K}$).

## Appendix B: Statistical methods

Estimates of the 95% confidence intervals are shown for the SEFDH calculations in Fig. 7. For ozone and water vapour in the SWOOSH dataset, a combined uncertainty arising from the uncertainties in the various instruments and a standard deviation arising from interannual variability can be obtained. These two quantities are of similar magnitude in the region of interest and a 95% confidence interval is obtained for each month by summing these two uncertainties in quadrature and assuming that each year in the dataset is independent. This assumption has been checked and is adequate. The SEFDH calculation for each constituent is then repeated to give bounds for the temperature change given the uncertainty in that constituent only. For example, the water vapour uncertainty in Fig. 7 is small and only reflects that coming from the water vapour dataset and not from differences in ozone (of about $\pm 0.05\,\mathrm{ppmv}$) which will also affect the temperature change from water vapour. However, the combined effect of both uncertainties is present in the calculation of the temperature change from both ozone and water





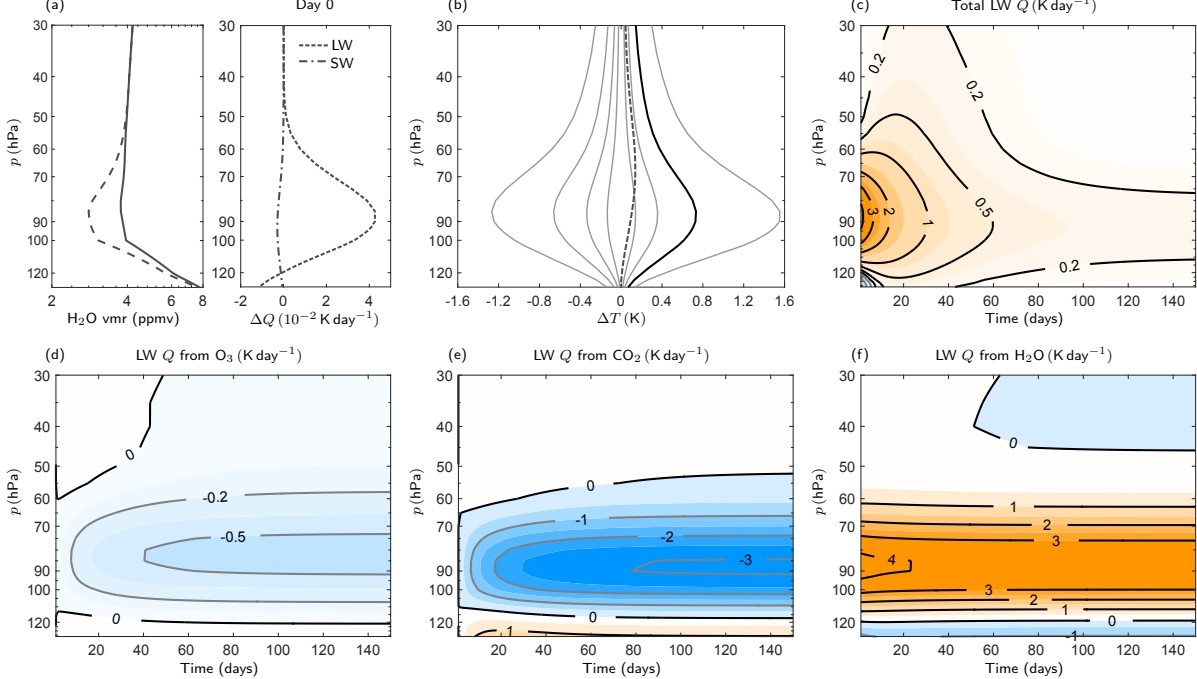

**Figure 14.** Similar to the plots in Figure 13 but for water vapour. (a) Left: water vapour profile (solid) and perturbation (dashed) used in the FDH calculation. Right: instantaneous change in heating rate from perturbation. (b) Temperature change resulting from the water vapour perturbation for $B_0 = 2.0, 1.0, 0.5, 0.2, -0.2, -0.5,$ **-1.0** (thick black line) and $-2.0$ ppmv from left to right at the maximum temperature change for the solid lines. For comparison, a perturbation in water vapour of a similar form to the ozone perturbation with $A_0 = 0.2$ ppmv is also included (dashed grey line) (see text for more details). For the perturbation $B_0 = -1.0$ ppmv, (c) is total longwave heating rate from all the constituents, (d) is longwave heating rate due to ozone, (e) is longwave heating rate due to carbon dioxide and (f) is longwave heating rate due to water vapour.

vapour. When calculating the peak to peak amplitude, the uncertainties at the maximum amplitude and minimum amplitude are added in quadrature.

The residual mean vertical velocity in reanalysis datasets has a large interannual variability and this is the only source of uncertainty taken into account in the calculation in Fig. 7. Again, in estimating this quantity, we assume that each year of the dataset is independent. This leads to a peak to peak amplitude from the dynamical heating averaged over $20°$ N–S at $70$ hPa of $5.6 \pm 0.6$ K and of $1.5 \pm 0.6$ K at $90$ hPa. In addition, there are other large discrepancies in estimates of the dynamical heating which are not taken into account in this calculation. For example, the difference between calculating the dynamical heating directly from $\overline{w}^* \, \overline{S}$ and from the thermodynamic equation can be as high as about 40% in certain months. A full treatment of all the sources of uncertainty in this calculation is beyond the scope of this work.





*Acknowledgements.* AM, ACM and PH were supported by an ERC ACCI grant (project no. 267760). In addition, ACM was supported by an AXA Postdoctoral Research Fellowship, and a NERC Research Fellowship (grant NE/M018199/1) and PH was supported by a NSERC postdoctoral fellowship. The authors wish to thank Stephan Fueglistaler for helpful discussions and Manoj Joshi for assistance with the IGCM3.1 model.



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
