# Peer review of "The radiative role of ozone and water vapour in the temperature annual cycle in the tropical tropopause layer"

_Atmospheric Chemistry and Physics, 2016_

## Referee Comment (RC1) · S. Solomon (Referee) · 18 Dec 2016

Major comments

1) a) What was assumed in this study regarding clouds? Are the results sensitive to clouds, both in the region from 100-130 mbar and below that level? b) How sensitive are the conclusions regarding the role of water vapor in the 100-130 mbar region to the background values of water vapor adopted, both in that region and below? The paper should discuss sensitivities to both the assumed water vapor background values and to cloudiness. If either or both are important, then what does that imply for the robustness

of the results presented?

2) The paper is quite long. It would greatly benefit from shortening with a particular eye to focusing on what is new here, and how robust it is. Results that are not robust to uncertainties should be edited, including where they are used to highlight differences relative to other studies. For example, the heavy emphasis on differences in ozone responses compared to Fueglistaler et al. on page 9 isn't warranted given the strong sensitivity to ozone climatology later stated on page 10. If the results are that sensitive, this doesn't merit a page of discussion. This occurs in several other places, and the paper would benefit from tightening throughout.

3) In many places, the review of past literature that is not new here could usefully be shortened. To give one example among many, potential reasons advanced in prior work as to why ozone varies between hemispheres don't need to be discussed in detail. This work is not about the reasons for ozone variations. Its focus is on their radiative effects.

4) Differences in the "smoothness" between SEFDH calculations and a 2-D dynamical model are heavily emphasized, but the computed changes are mainly in a few limited regions. It is useful and very helpful that the authors show that there is essentially no difference in the tropical mean. But it's also true that over much of the region from 20N-20S, Figures 10a,b and 11 show differences of less than 20% between the SEFDH and 2-D dynamical model calculations; i.e., Figures 10 and 11 show that the SEFDH and 2-D model agree quite well in more places and times than they disagree. Highlighting local differences that occur only in limited places doesn't provide a balanced representation of the findings. Language should be changed throughout the paper to avoid over-emphasizing spatially limited changes, and a more balanced account should be provided.

5) a) Are your statements about differences in the SEFDH and 2D calculations robust to uncertainties in the adopted constituent distributions? Given the strong sensitivity of

the results to the background climatologies in ozone, how might errors in the SWOOSH dataset's background climatologies of ozone as a function of latitude affect your results on this point? What about water vapor gradients? I would expect dynamical responses to a radiative perturbation to depend upon the background climatological gradients and was surprised that there was no discussion of that.

6) The paper does not present a clear case for what causes the decrease in 'smoothness' for the SEFDH versus IGCM, which is the central key point in the paper. The comparison of changes in vertical velocity in Figure 10c and the difference between the SEFDH and IGCM calculations suggests as many mismatches as it does matches, so this on its own doesn't serve to convince the reader. It is suggested in the text on the bottom of page 22, top of page 23 that there is a balance involving Qrad, the time rate of change of temperature, and dynamical heating, but the paper doesn't demonstrate a balance. To be publishable, the paper needs to show exactly how these (or other) factors change in the model to produce the results shown. It would be appropriate to do that for a few of the key places where there are larger local differences between the SEFDH and the IGCM, and a few of the places where there are no such large changes.

7) It would be helpful to have more information here on how the 2-D dynamical model performs. 2-D models obviously have many limitations, and the references cited generally focus more on broad dynamical phenomena than on quantitative performance. Does the 2-D model generate accurate seasonal and latitudinal climatologies of temperature, winds, and circulation from apriori information? Is the model mean circulation or background temperature distribution tuned? How much confidence is there in the model's ability to simulate the strength of the Brewer-Dobson circulation (critical here), and how has it been tested? More discussion of confidence in the model's quantitative performance is needed, since the paper's key findings rest on a robust quantitative simulation of meridional circulation perturbations from a model of reduced dimensionality.

8) The paper doesn't state what years were used to define the background climatologies for H2O and O3 against which seasonal anomalies were evaluated from among

the range of 1984-2015 available in SWOOSH. This is particularly concerning for ozone, where it is clear that there have been long-term trends in the tropics. Decadal variability in tropical H2O has also been established in the scientific literature. What years were used? How much do the years chosen matter, both in terms of background climatologies and amplitudes of the responses you are interested in, both for ozone and water vapor?

Detailed comments

1, Lines 8-9. Given the high sensitivity to the adopted background ozone values, as you show in the text, it isn't clear that your results being high deserves such emphasis here.

1, Lines 12-13. Is the non-local result robust to assumptions about background water vapor and background and seasonality of cloudiness? Clarify or delete.

4, Lines 10-13. Should tropical gravity waves be explicitly noted as a possible factor in tropical upwelling?

6, Lines 18-19. What is meant by "a three-point Gaussian is used to account for the diurnal variation in solar zenith angle"? Please clarify what you assume regarding diurnal changes in heating rates.

7, Lines 1-2. Are you sure that the temperature changes would necessarily be zonally uniform if, for example, there are zonal asymmetries in potentially thick clouds?

9, Lines 21-23. Why is it necessary to presume this rather than determining whether this is actually true in your calculations?

Page 10, line 13. The claim of a 10% accuracy in the SWOOSH dataset is remarkable. Is this 2-sigma? Does it apply for local values across the full range of latitudes of interest here, all the way down to 130 mbar? What is this claim based on? I would not expect 2-sigma absolute uncertainties in tropical ozone in SWOOSH to be better than 20 or 30%, at best.

Page 11, line 17. Why 'perhaps'? Please provide a clear statement based on your quantitative results as to whether it is or isn't.

Page 13, line 8. Garbled sentence.

Page 14, line 18. A secondary

Page 20, line 2. Why 'maximum possible relevance...to the real atmosphere'? A full 3-D model, and a complete line-by-line radiative code would have more relevance to the real world, so this statement should be deleted.

Sections 5.2, 5.3, 5.4, and 6 require revision to deal with the above comments. I will not repeat those remarks here on a line-by-line basis but they occur in many places.

Page 25, lines 2-4. Gilford and Solomon's paper has been accepted in J. Climate. Your statement that your consideration of different vertical layers, and water vapor, is different from that paper is not correct. Gilford and Solomon did consider water vapor, as well as concentration perturbations in different layers. Please revise to accurately quote what Gilford and Solomon did.

Page 28, lines 15-21. 'should not be taken too seriously' is not clear and it's not scientific language. Please provide a quantitative statement that is specific, and balanced across regions of agreement and disagreement and considers robustness of your results as noted above in the major comments section.

---

## Referee Comment (RC2) · Anonymous Referee #1 · 24 Dec 2016

**General**

Ming et al. analyze the role of radiatively active trace gas variations on the amplitude, phase and structure of the annual cycle in lower stratospheric temperatures through their impact on the radiative energy budget. The tracer variations are understood to be a consequence of the same (dynamical) process that drives the temperature variations. In line with previous authors, they find that ozone strongly amplifies the temperature amplitude, while water vapor dampens the temperature amplitude. The work presented is a major step forward - related previous work has mainly focussed on making the case that these "thermal forcings" are important, and cannot be neglected. The present work presents a much more quantitative analysis and, importantly, integrates

the thermal forcing in a (simplified) general circulation model, which shows (not unexpectedly) that the assumption underlying SEFDH (vertical velocity remains constant) is not well justified. The paper is well written, but could be shortened in some place. I have mostly technical comments, listed below.

**Minor comments**

Abstract: I think the abstract could be somewhat condensed, and (i) more quantitative information could be given; (ii) the text switches back and forth between amplitude, phase, ozone, water and temperature (e.g. around Lines 6-12), which is confusing.

P1/L7,8: "... and out of phase with the observed annual cycle of the tracer mixing ratio."

P1/L8: Suggest: "The ozone contribution calculated here is ..."

P1/L9: Suggest to replace "This difference" with "The difference"; but see general comment on Abstract above.

P2/L20: Suggest (the tropopause temperature is important for stratospheric water, not dehydration per se.) "Therefore the temperatures at 90hPa and 100hPa shown in Figs 1(c/d) are more directly relevant to water entering the stratosphere than the temperature at 70hPa."

P4/L26: Quantitative details for H2O missing in previous work. Fueglistaler et al. state that their calculations " show that the temperature adjustment for the annual cycle in water vapour around 70 hPa is about an order of magnitude smaller than that for ozone." At the time of writing Fueglistaler et al., this seemed to be sufficient information. However, in the context of your ozone effect being substantially larger than theirs, it is interesting to note that their water vapor impact (around 0.2K) is also smaller than your water vapor impact (your Figure 5b gives 0.6K).

I think it would be great if your paper could resolve these quantitative differences better, such that your paper can be cited as the best quantification of the radiative effects. The Fueglistaler et al. numbers are taken from the calculations with the Edwards-

Slingo code, but results with the Fu-Liou code were rather similar (see Fueglistaler et al., page 3702; last paragraph). The calculations using the Fu-Liou code gave slightly larger amplitudes, but the differences were much smaller than the differences to your calculations; at the time, we decided to go with the Edwards-Slingo results as they were obtained from a software configuration that has been used in previous studies by Piers Foster. It would be great if you could resolve the questions you discuss on page 10/11 regarding the role of the radiative transfer code versus trace input data - the ozone and water vapor files used by Fueglistaler et al. for the calculations with the Edwards-Slingo and Fu-Liou code can be made available.

Finally, can you quantify the impact on the global average temperature? An important point of Fueglistaler et al. is that one should not assume that the global average temperature remains constant (as e.g. implied in Yulaeve et al.1994); it would be interesting to see the numbers from your calculation. You do discuss the role of extratropical heating rate variations on the tropical amplitude, and it appears to me it that would be fairly straightforward to also show the annual mean cycle of global average temperatures.

---

## Author Comment (AC1) · 6 Mar 2017

**Response to reviewers**

Alison Ming, Amanda Maycock, Peter Hitchcock and Peter Haynes

March 6, 2017

We would like to thank both reviewers for their detailed comments which have helped us to improve the paper significantly in revision. We have implemented many of the suggestions and provided detailed responses when we have not. In light of the comments from both reviewers about the length of the manuscript, we have revised the text to make the paper more concise.

Page and line numbers quoted in the responses refer to the updated text.

In response to the comments of Reviewer 1 concerning the differences between SEFDH and dynamical calculations we have reexamined the IGCM calculation and modified it slightly to minimise any differences between it and the SEFDH calculation arising from implementation rather than dynamical adjustment.

Below we indicate how we have dealt with each individual comment.

**Reviewer 1**

What was assumed in this study regarding clouds? Are the results sensitive to clouds, both in the region from 100-130 mbar and below that level?

All calculations in the paper assume clear-sky conditions as stated on page 4, line 15. While it is possible that cloud effects could modify the TTL temperature responses to ozone and water vapour, and as such a careful exploration of this effect would be a valuable contribution, we note that our calculations accounting for the effects of both clear-sky radiative and dynamical heating reproduce well the observed annual cycle in TTL temperature, suggesting that the net effects of clouds (other than those included indirectly in the dynamical heating) are small, justifying this choice of focus.

Nevertheless, in revising the paper we have thought carefully about cloud effects and have performed additional SEFDH calculations to give an indication of the order of magnitude of cloud effects. These calculations show the effect of including annual mean climatological cloud cover is to decrease the peak-to-peak annual cycle temperature change due to ozone at $70\,\mathrm{hPa}$ by 5-10% at all latitudes between $20°\,\mathrm{N}$ and $20°\,\mathrm{S}$. The effect on the water vapour annual cycle at the same level is negligible. For this estimate, we used the high, medium and low annual mean cloud fractions from the ISCCP dataset (1983 to 2006). The clouds lead primarily to a reduction in the amount of upwelling longwave radiation reaching $70\,\mathrm{hPa}$ of about $0.05\,\mathrm{K\,day^{-1}}$ which in turn decreases the ozone temperature response. Clouds also have an annual cycle in radiative heating. One estimate, at a single location in the tropics, from Fueglistaler and Fu (2006) shows that the annual cycle in radiative heating from clouds is about $0.05\,\mathrm{K\,day^{-1}}$ (peak to peak) at $70\,\mathrm{hPa}$ but these values are at present poorly constrained. On the basis of these rough quantitative estimates we consider it unlikely that including cloud effects will invalidate our main arguments and conclusions.

However, we do not feel that it is practical or appropriate to present these estimates in the revised version of the paper – the problem is sufficiently complicated that a brief explanation could not do it justice and the scientific uncertainties are large.

> b) How sensitive are the conclusions regarding the role of water vapor in the 100-130 mbar region to the background values of water vapor adopted, both in that region and below? The paper should discuss sensitivities to both the assumed water vapor background values and to cloudiness. If either or both are important, then what does that imply for the robustness of the results presented?

The results are not sensitive to the background concentrations of water vapour within the ranges of values in the SWOOSH dataset (page 16 Line 1-3). The main sensitivity is to background concentrations of ozone which we have now discussed in more detail with supporting calculations added in Appendix C.

> 2) The paper is quite long. It would greatly benefit from shortening with a particular eye to focusing on what is new here, and how robust it is. Results that are not robust to uncertainties should be edited, including where they are used to highlight differences relative to other studies. For example, the heavy emphasis on differences in ozone responses compared to Fueglistaler et al. on page 9 isnt warranted given the strong sensitivity to ozone climatology later stated on page 10. If the results are that sensitive, this doesnt merit a page of discussion. This occurs in several other places, and the paper would benefit from tightening throughout.

The paper has been considerably shortened. The main text (excluding references and appendices) has been reduced from 28 pages to 24 pages during revision. In particular, the review of current literature in the introduction has been made more concise and the discussion of the differences compared to Fueglistaler et al. (2011) is now shorter.

> 3) In many places, the review of past literature that is not new here could usefully be shortened. To give one example among many, potential reasons advanced in prior work as to why ozone varies between hemispheres dont need to be discussed in detail. This work is not about the reasons for ozone variations. Its focus is on their radiative effects.

This point has been addressed together with the comment above whilst shortening the paper.

4) Differences in the "smoothness" between SEFDH calculations and a 2-D dynamical model are heavily emphasized, but the computed changes are mainly in a few limited regions. It is useful and very helpful that the authors show that there is essentially no difference in the tropical mean. But its also true that over much of the region from 20N- 20S, Figures 10a,b and 11 show differences of less than 20% between the SEFDH and 2-D dynamical model calculations; i.e., Figures 10 and 11 show that the SEFDH and 2-D model agree quite well in more places and times than they disagree. Highlighting local differences that occur only in limited places doesnt provide a balanced representation of the findings. Language should be changed throughout the paper to avoid over-emphasizing spatially limited changes, and a more balanced account should be provided.

We have added Figure 10(c) which shows explicitly the difference between the SEFDH and dynamical temperature responses.

The validity of assuming fixed dynamical heating within the tropics has been questioned by a number of important studies on this topic. Fels et al. (1980): "Supporting auxiliary calculations using purely radiative models are also presented. One of these, in which the thermal sensitivity is computed using the assumption that heating by dynamical processes is unaffected by changed composition, gives results that are generally in excellent agreement with the GCM. Exceptions to this occur in the ozone reduction experiment at the tropical tropopause..."– i.e., Fels et al. (1980) are saying that (SE)FDH may give a poor estimate of the temperature response in the tropical tropopause region to changes in ozone in this region. Garcia (1987): "This implies that tropical heating will tend to be balanced by a mean meridional circulation rather than by radiative relaxation, in agreement with the conclusions of Fels et al. (1980)." If this statement by Garcia (1987) was taken as a rule of thumb then it would imply that a zeroth-order approach to considering the effect of changing ozone in the tropical tropopause region should be to assume that all the anomalous heating is balanced by a change in dynamical heating – i.e. a kind of anti-SEFDH. In practice, the extent to which the temperature response to an ozone perturbation is correctly predicted by SEFDH comes down to details, in particular the latitudinal structure in the ozone perturbation. The heating anomalies associated with broad features (more than 20 degrees of latitude for this problem, as argued in the paper) give rise to a broad temperature change as predicted by SEFDH. The heating anomalies associated with narrow features are primary balanced by changes in dynamical heating and the temperature response is small. So simply giving the size of local differences does not seem an effective description of how things work.

We believe that this characterization in terms of latitudinal scale is the most useful and general way to describe the effectiveness (or otherwise) of the SEFDH calculation, rather than focusing on whether or not numerical values are in agreement in particular locations. Nonetheless, in revision we have taken account of this comment from the reviewer by trying to ensure that our description is as clear and quantitative as possible (see also Point 6 below). As such, we have explicitly shown the differences between the SEFDH and dynamical adjustment calculations.

5) a) Are your statements about differences in the SEFDH and 2D calcula-
tions robust to uncertainties in the adopted constituent distributions? Given the
strong sensitivity of the results to the background climatologies in ozone, how
might errors in the SWOOSH datasets background climatologies of ozone as
a function of latitude affect your results on this point? What about water vapor
gradients? I would expect dynamical responses to a radiative perturbation to
depend upon the background climatological gradients and was surprised that
there was no discussion of that.

As explained in the paper and noted above, the differences in SEFDH and 2D dynamical
calculations are largest when latitudinal gradients in the anomalous heating implied by the
change in constituents are large. Therefore, broadly speaking, errors in the SWOOSH dataset
on small horizontal scales will have a greater effect on the differences (and will be manifested
in small horizontal scales), while errors on larger scales will have a less significant effect on the
difference.

In a similar vein, the effect of errors in the background climatologies will largely be determined
by how those errors modulate the anomalous heating associated with the annual variations in
ozone and water vapour. If errors in the climatologies tend to reduce the small horizontal scale
features in that anomalous heating, then the corresponding SEFDH vs. dynamical differences
will be smaller. If background errors tend to increase the small horizontal scale features, then
the differences will be larger. There does not seem to be any particular reason why errors in the
background field should systematically reduce or increase the small horizontal scale features in
the anomalous heating. Furthermore, irrespective of the effect of such errors, the same broad
principle applies – large horizontal scale features in SEFDH calculated temperature response
are likely to be robust, small horizontal scale features are not.

6) The paper does not present a clear case for what causes the decrease in
'smoothness' for the SEFDH versus IGCM, which is the central key point in the
paper. The comparison of changes in vertical velocity in Figure 10c and the
difference between the SEFDH and IGCM calculations suggests as many mis-
matches as it does matches, so this on its own doesnt serve to convince the
reader. It is suggested in the text on the bottom of page 22, top of page 23 that
there is a balance involving Qrad, the time rate of change of temperature, and
dynamical heating, but the paper doesnt demonstrate a balance. To be pub-
lishable, the paper needs to show exactly how these (or other) factors change
in the model to produce the results shown. It would be appropriate to do that
for a few of the key places where there are larger local differences between the
SEFDH and the IGCM, and a few of the places where there are no such large
changes.

In addressing this important comment (some aspects of which are addressed by our replies above to 4 and 5), we have first made some minor modifications to the implementation of the radiative code in the IGCM calculation to ensure that the SEFDH and IGCM calculations are directly comparable. These changes have only a minor effect in the updated Figure 10 (see below), but this verifies that differences between the SEFDH and dynamical adjustment calculations do not arise simply from differences in implementation.

The following text has been added:

[Figure]

Figure 10: (a) Monthly temperature changes showing the annual cycle at 70 hPa calculated using the idealised dynamical model (IGCM) with an annual cycle in ozone. (b) Figure 3(b) is reproduced here for comparison and shows the corresponding SEFDH calculation at 70 hPa. (c) Difference in temperature change (K) between SEFDH calculation, (b), and the IGCM calculation,(a). (d) Change in upwelling in idealised dynamical model. (e) Temperature change at 70 hPa calculated by imposing the term $\Delta(\overline{w}\,\overline{S})$ from the dynamical model as a perturbation to the SEFDH calculation. See main text for more details.

"Figures 10(a) and (b) compare the temperature change at 70 hPa caused by the annual cycle in ozone in the dynamical model and in the SEFDH calculation, respectively (Fig. 10(b) is identical to Fig. 3(b), but is included here for ease of comparison). Fig. 10(c) shows the difference between the two. The figures show the importance of including the dynamical adjustment, which tends to broaden the temperature response in latitude in the tropical region, making it more symmetric about the Equator. Note, in particular, the effect on the off-equatorial maximum at about $10°$ N in the SEFDH calculation, which is no longer a distinct isolated feature in the dynamical calculation."

We have further added Figure 10(e), which shows the temperature response induced by the dynamical heating from the change in upwelling shown in Figure 10(d). The temperature response

in Figure 10(e) closely matches the SEFDH - dynamical adjustment temperature differences in Figure 10(c), thus demonstrating quantitatively that the temperature differences are explained by the dynamical response. We then include a discussion of the criterion that determines the range of latitudes over which dynamical heating response will dominate.

7) It would be helpful to have more information here on how the 2-D dynamical model performs. 2-D models obviously have many limitations, and the references cited generally focus more on broad dynamical phenomena than on quantitative performance. Does the 2-D model generate accurate seasonal and latitudinal climatologies of temperature, winds, and circulation from apriori information? Is the model mean circulation or background temperature distribution tuned? How much confidence is there in the models ability to simulate the strength of the Brewer-Dobson circulation (critical here), and how has it been tested? More discussion of confidence in the models quantitative performance is needed, since the papers key findings rest on a robust quantitative simulation of meridional circulation perturbations from a model of reduced dimensionality.

The dynamical model is being used to calculate responses to the changes in constituents, not calculate 'background quantities such as the time-average Brewer-Dobson circulation. There is no sense in which the dynamical model is tuned. It simply contains the basic ingredients of the 2-D dynamical equations, plus radiative relaxation as represented by a radiative calculation considering the temperature response as an anomaly relative to a specified background temperature field. We noted in the original text that tests had shown that the calculated response was insensitive to plausible changes in that background temperature field – e.g., changing it from annual average to solstice distributions.

We have made it clear that the dynamical model does not include the effects of change in eddy forcing and we have carefully discussed the limitations of that.

8) The paper doesnt state what years were used to define the background climatologies for H2O and O3 against which seasonal anomalies were evaluated from among the range of 1984-2015 available in SWOOSH. This is particularly concerning for ozone, where it is clear that there have been long-term trends in the tropics. Decadal variability in tropical H2O has also been established in the scientific literature. What years were used? How much do the years chosen matter, both in terms of background climatologies and amplitudes of the responses you are interested in, both for ozone and water vapor?

This information is given on page 5 line 13. The seasonal cycle is provided by SWOOSH using data from 1984 to 2015. The interannual variability is incorporated into the uncertainty estimates for both ozone and water vapour (see Appendix B).

Detailed comments: 1, Lines 8-9. Given the high sensitivity to the adopted background ozone values, as you show in the text, it isnt clear that your results being high deserves such emphasis here.

We were intending here to put our results into the context of other literature on this topic. However, taking our results together with those of Gilford and Solomon (2017) and Chae and Sherwood (2007), all of which show a larger amplitude for the ozone annual cycle effect than Fueglistaler et al. (2011), we have removed this clause from the abstract. A discussion of the quantitative magnitudes and how these compare to other literature is still given in the main text (Pages 6 to 9).

1, Lines 12-13. Is the non-local result robust to assumptions about background water vapor and background and seasonality of cloudiness? Clarify or delete. tropical upwelling?

The non-local radiative result arises from the properties of the radiative transfer for water vapour in the TTL and is robust to background water vapour [tested and reported in the paper, page 11, line 5] and tropospheric cloudiness [tested and not explicitly reported in the paper]. Neither the effects of background cirrus nor the seasonality in cloudiness have been considered, but this is clearly stated as outside the scope of the paper.

4, Lines 10-13. Should tropical gravity waves be explicitly noted as a possible factor in tropical upwelling?

Tropical gravity waves are possibly relevant but whilst condensing the review of past literature, as recommended by the referee, this paragraph has been modified to

"Many studies have focused on the role of wave-induced forces in driving the annual cycle in temperature through their effects on upwelling in the TTL and hence on $\overline{Q}_{\text{dyn}}$ in Eq. 1, although uncertainty remains about what types of waves are the most important (Randel and Jensen, 2013, and references therein)."

6, Lines 18-19. What is meant by "a three-point Gaussian is used to account for the diurnal variation in solar zenith angle"? Please clarify what you assume regarding diurnal changes in heating rates.

The Morcrette/Zhong and Haigh radiation code incorporates a common method of calculating a diurnal average which accounts for the variation in solar zenith angle over the day. Note that how we compute these diurnal averages does not affect the results in this paper so we have revised this sentence to "Shortwave heating rates are calculated as diurnal averages".

7, Lines 1-2. Are you sure that the temperature changes would necessarily be zonally uniform if, for example, there are zonal asymmetries in potentially thick clouds?

This is a statement about our calculation, in which all quantities are assumed to be independent of longitude, rather than a statement that 'if TTL temperatures and constituents are zonally uniform then the temperature response will be zonally uniform'. We have amended the text to clarify this as well as to emphasise the fact that the zonal mean adjustment would smooth regional changes in temperature that would be predicted by an SEFDH calculation.

9, Lines 21-23. Why is it necessary to presume this rather than determining whether this is actually true in your calculations?

This sentence has been edited to remove the word "presumably". This claim is essentially verified by Fig. 3; the sentence has accordingly been strengthened.

Page 10, line 13. The claim of a 10% accuracy in the SWOOSH dataset is remarkable. Is this 2-sigma? Does it apply for local values across the full range of latitudes of interest here, all the way down to 130 mbar? What is this claim based on? I would not expect 2-sigma absolute uncertainties in tropical ozone in SWOOSH to be better than 20 or 30%, at best.

A consideration of the uncertainties as far as they have been characterized for the SWOOSH dataset (see Davis et al., 2016) suggests that 10% for climatological values is reasonable at $70\,\mathrm{hPa}$

although this uncertainty increases to 30-40% by 130 hPa. We emphasise to the referee that these calculations are included in the manuscript to highlight the sensitivity of the calculations to the background ozone profile, and as such the 10% values quoted are intended to represent plausible ranges of ozone values rather than a precise assessment of uncertainty, for which are more detailed explanation for the SWOOSH dataset is given in Davis et al. (2016).

We have also added an Appendix C with additional calculations to describe the effect of varying background ozone values which is the primary source of sensitivity in our results. These calculations are intended as illustrative calculations rather than a precise assessment of uncertainty.

> Page 11, line 17. Why 'perhaps'? Please provide a clear statement based on your quantitative results as to whether it is or isnt.

The word 'perhaps' has been removed.

> Page 13, line 8. Garbled sentence.

This sentence was too long and has been revised to:

"In each case, the latitude of the maximum response is further north than the latitude of the maximum amplitude in the water vapour mixing ratios at that level. The fact that there is no simple relation between the latitude-time structure of the SEFDH-predicted annual cycle in temperature at a given level and the latitude-time structure of the water vapour mixing ratios at that level is further evidence for important non-local contributions in the vertical from water vapour to the temperature variations."

> Page 14, line 18. A secondary

It is not clear from this comment what the reviewer's concern here was; however the relevant text has been modified in the course of shortening the paper.

> Page 20, line 2. Why 'maximum possible relevance ... to the real atmosphere'? A full 3-D model, and a complete line-by-line radiative code would have more relevance to the real world, so this statement should be deleted.

The text has been amended.

Sections 5.2, 5.3, 5.4, and 6 require revision to deal with the above comments. I will not repeat those remarks here on a line-by-line basis but they occur in many places.

These Sections have been revised. See responses to previous comments.

Page 25, lines 2-4. Gilford and Solomons paper has been accepted in J. Climate. Your statement that your consideration of different vertical layers, and water vapor, is different from that paper is not correct. Gilford and Solomon did consider water vapor, as well as concentration perturbations in different layers. Please revise to accurately quote what Gilford and Solomon did.

We have modified our discussion of Gilford and Solomon (2017) accordingly.

Page 28, lines 15-21. 'should not be taken too seriously' is not clear and its not scientific language. Please provide a quantitative statement that is specific, and balanced across regions of agreement and disagreement and considers robustness of your results as noted above in the major comments section.

The text has been amended to improve clarity and exploit the new Fig. 10. As noted in earlier replies, the key and robust point about the difference between SEFDH temperature response and dynamical temperature response is that this is largest at small horizontal scales. As can be seen in the revised Fig. 10(c), local differences are zero for some latitudes and dates but for others, these are up to about $1.5\,\mathrm{K}$, i.e., about 50% of the maximum SEFDH 'signal'. Again, simply giving the size of local differences does not seem an effective description of how things work.

**Bibliography**

Chae, J. H., and S. C. Sherwood, 2007: Annual temperature cycle of the tropical tropopause: A simple model study. *Journal of Geophysical Research*, **112 (D19)**, D19 111, doi:10.1029/2006JD007956, URL http://doi.wiley.com/10.1029/2006JD007956.

Fels, S. B., J. D. Mahlman, M. D. Schwarzkopf, and R. W. Sinclair, 1980: Stratospheric Sensitivity to Perturbations in Ozone and Carbon Dioxide: Radiative and Dynamical Response. *Journal of the Atmospheric Sciences*, **37 (10)**, 2265–2297, doi:10.1175/1520-0469(1980)037⟨2265: SSTPIO⟩2.0.CO;2.

Fueglistaler, S., and Q. Fu, 2006: Impact of clouds on radiative heating rates in the tropical lower stratosphere. *Journal of Geophysical Research Atmospheres*, **111 (23)**, 1–13, doi:10.1029/2006JD007273.

Fueglistaler, S., P. H. Haynes, and P. M. Forster, 2011: The annual cycle in lower stratospheric temperatures revisited. *Atmospheric Chemistry and Physics*, **11 (8)**, 3701–3711, doi:10.5194/acp-11-3701-2011, URL http://www.atmos-chem-phys.net/11/3701/2011/.

Garcia, R. R., 1987: On the Mean Meridional Circulation of the Middle Atmosphere. *Journal of Atmospheric Sciences*, **44 (24)**, 3599–3609, doi:10.1175/1520-0469(1987)044⟨3599: OTMMCO⟩2.0.CO;2.

Gilford, D. M., and S. Solomon, 2017: Radiative effects of stratospheric seasonal cycles in the tropical upper troposphere and lower stratosphere. *Journal of Climate*, JCLI–D–16–0633.1, doi:10.1175/JCLI-D-16-0633.1, URL http://journals.ametsoc.org/doi/10.1175/JCLI-D-16-0633.1.

Randel, W. J., and E. J. Jensen, 2013: Physical processes in the tropical tropopause layer and their roles in a changing climate. *Nature Geoscience*, **6 (3)**, 169–176, doi:10.1038/ngeo1733, URL http://www.nature.com/doifinder/10.1038/ngeo1733.

---

## Author Comment (AC2) · 6 Mar 2017

**Response to reviewers**

Alison Ming, Amanda Maycock, Peter Hitchcock and Peter Haynes

March 6, 2017

We would like to thank both reviewers for their detailed comments which have helped us to improve the paper significantly in revision. We have implemented many of the suggestions and provided detailed responses when we have not. In light of the comments from both reviewers about the length of the manuscript, we have revised the text to make the paper more concise.

Page and line numbers quoted in the responses refer to the updated text.

Below we indicate how we have dealt with each individual comment.

**Reviewer 2**

> Abstract: I think the abstract could be somewhat condensed, and (i) more quantitative information could be given; (ii) the text switches back and forth between amplitude, phase, ozone, water and temperature (e.g. around Lines 6-12), which is confusing.

The abstract has been revised to discuss the ozone annual cycle first then the water vapour.

> P1/L7,8: "... and out of phase with the observed annual cycle of the tracer mixing ratio."
> P1/L8: Suggest: "The ozone contribution calculated here is ..."
> P1/L9: Suggest to replace "This difference" with "The difference"; but see general comment on Abstract above.

These suggestions have been taken into account in the revised text.

> P2/L20: Suggest (the tropopause temperature is important for stratospheric water, not dehydration per se.) "Therefore the temperatures at 90hPa and 100hPa shown in Figs 1(c/d) are more directly relevant to water entering the stratosphere than the temperature at 70hPa."

This is a good point and the sentence has been changed.

P4/L26: Quantitative details for H2O missing in previous work. Fueglistaler et al. state that their calculations "show that the temperature adjustment for the annual cycle in water vapour around 70 hPa is about an order of magnitude smaller than that for ozone." At the time of writing Fueglistaler et al., this seemed to be sufficient information. However, in the context of your ozone effect being substantially larger than theirs, it is interesting to note that their water vapor impact (around 0.2K) is also smaller than your water vapor impact (your Figure 5b gives 0.6K). I think it would be great if your paper could resolve these quantitative differences better, such that your paper can be cited as the best quantification of the radiative effects. The Fueglistaler et al. numbers are taken from the calculations with the Edwards-Slingo code, but results with the Fu-Liou code were rather similar (see Fueglistaler et al., page 3702; last paragraph). The calculations using the Fu-Liou code gave slightly larger amplitudes, but the differences were much smaller than the differences to your calculations; at the time, we decided to go with the Edwards-Slingo results as they were obtained from a software configuration that has been used in previous studies by Piers Foster. It would be great if you could resolve the questions you discuss on page 10/11 regarding the role of the radiative transfer code versus trace input data -the ozone and water vapor files used by Fueglistaler et al. for the calculations with the Edwards-Slingo and Fu-Liou code can be made available.

We have performed a number of SEFDH calculations that try to narrow down the causes of the differences between our calculation and Fueglistaler et al. and whilst we have not been able to explain the differences, the main effect is likely to be differing ozone climatologies rather than radiation code. The temperature change was more sensitive to the background ozone field than any other changes we tested. For instance, an SEFDH calculation with the RRTM radiation code and SWOOSH ozone obtained similar results to those in this manuscript.

We suggest that future studies, as far as possible, make the actual fields of ozone and water vapour used in the radiative calculation publicly available, to make comparison more straightforward.

Finally, can you quantify the impact on the global average temperature? An important point of Fueglistaler et al. is that one should not assume that the global average temperature remains constant (as e.g. implied in Yulaeve et al. 1994); it would be interesting to see the numbers from your calculation. You do discuss the role of extratropical heating rate variations on the tropical amplitude, and it appears to me it that would be fairly straightforward to also show the annual mean cycle of global average temperatures.

The plot below shows the global averaged temperature in the IGCM model calculation and ERA-Interim. This supports the point that the global average temperature is not constant with a peak to peak amplitude of about 1.4 K in the case of the ozone and water vapour perturbation. The 20° N-S averaged for this calculation was about 2.8 K.

[Figure]

Figure 1: Globally averaged temperature at 70 hPa in the IGCM model calculations with ozone and water vapour perturbations. The ERA-Interim plot is also shown.

We have considered including a mention of the global average but having significantly reduced the length of the paper, there did not seem an appropriate location for the detailed discussion that this topic would require.

---

## Author Response (AR2)

**Response to reviewers**

Alison Ming, Amanda Maycock, Peter Hitchcock and Peter Haynes

April 7, 2017

We are grateful to the reviewer for the additional comments and suggestions. We have revised the wording of the text in many places to be more precise and have addressed the various comments as follows:

> 1) In the response document, there are now statements about the IGCM calculation having potential differences 'arising from implementation rather than dynamical adjustment'. This was not mentioned in the original manuscript. What causes potential differences in 'implementation? Does this remain a potential source of error in the revised draft, and how well quantified is it?

The differences in implementation were found after careful comparison of the SEFDH and IGCM calculations that was prompted by the reviewer's previous comments. We therefore made changes to the implementation of the radiation code so it was directly comparable to the SEFDH version. One such change was to calculate the diurnally averaged shortwave in the same way in the two calculations and another was to change the way in which the trace gas concentrations were provided to the radiation code in the IGCM. As we explained in our previous responses, Figs 10c and 10e were included in the revised version to demonstrate explicitly that the differences between the dynamical calculation and the SEFDH calculations could be explained by the effect of dynamical heating, rather than being the result of differences in the detailed implementation of the two calculations. There are small differences between the two figures but the overall agreement is very good. Therefore this does not remain a potential source of error.

The revised paper explained this and contained the sentence

"The resulting temperature response shown in Fig. 10(e) is a very good match to the difference in temperature in Fig. 10(c)."

– for clarity we have further modified this to (page 19, line 8)

"The resulting temperature response shown in Fig. 10(e) is a very good match to the difference in temperature in Fig. 10(c) and reassures us that the difference between SEFDH and dynamical calculations can indeed be interpreted as resulting from the effect of dynamical heating and is not due to differences of detail in the implementation of the two calculations."

Once again we are grateful to the reviewer for focusing our attention on this point in her comments on the ACPD version of the paper.

> 2) In the response document and in the manuscript, it is stated that since the results reproduce the annual cycle then the net effect of clouds must be small. The response document refers to 'rough quantitative estimates to back this up, but that is quite loose language. How rough? How quantitative? It is later

stated that no results regarding this concern will be given in the draft because 'the problem is sufficiently complicated that a brief explanation could not do it justice and the scientific uncertainties are large'. So the reviewer is first presented with an assertion that the effect of clouds must be a small one (because the seasonal cycle looks good without it), and second a statement that the uncertainties are large. This is not an appropriate response. Getting the 'right' answer for the annual cycle can happen for wrong reasons, and the assertion that clouds are not important even though uncertainties are large is speculation. Particularly given the potential for cloud feedbacks in a dynamical calculation to affect radiation (e.g., through changes in cirrus in regions of altered upwelling, although that is only one example among many), I think further information is required beyond the statements made that this is 'beyond the scope of this work'. The paper needs to be revised to address this more clearly.

We have removed the sentence that suggests "the net effect of clouds must be small". Instead we have added the following paragraph to the discussion to clearly state that a full quantification of the cloud effect will require further work (page 23, line 8):

"All of the calculations make use of a clear-sky assumption. A rough SEFDH calculation taking into account an estimate of the annual mean climatological high cloud cover shows that the peak-to-peak annual cycle temperature change due to ozone at $70\,\mathrm{hPa}$ decreases by 5-10% at all latitudes between $20^\circ\,\mathrm{N}$ and $20^\circ\,\mathrm{S}$. The effect on the water vapour annual cycle at the same level is negligible. The clouds lead primarily to a reduction in the amount of upwelling longwave radiation reaching $70\,\mathrm{hPa}$ of about $0.05\,\mathrm{K\,day}^{-1}$ which in turn decreases the ozone temperature response. A full assessment of the cloud effect is beyond the scope of this work and further work is needed to establish their precise contribution."

3) I did not find the authors response to my question 7) to be sufficient. I am asking about the mean meridional circulation in this representation of the model (as implemented, which could differ from other uses of the IGCM that may be discussed in the literature). Since the paper seeks to calculate thermal responses associated with imposed constituent changes, it seems to me to be quite important to determine whether or not the mean meridional circulation that is found in this representation of the model prior to imposing the perturbation is accurate. One approach that can be used is tracer transport tests, for example. I dont find the statement in the paper regarding the difference of 2K from a control run in the tropical stratosphere to be proof that the reader should be confident that an accurate balance exists in this representation of the IGCM model between radiation and vertical motion near the tropical tropopause. Both of these terms are known to be important drivers of temperature seasonal cycles in the region of interest, and 2K would be a very important change if it occurs close to the tropical tropopause. A 4 year long 'control run' is carried out and results for the perturbed versus control are shown for a 5th year, but the rationale for a 4-year

control is not given when it is introduced, and there is no discussion of how the model behaves over the 'spin-up' period in this configuration compared to IGCM 3.1. If the spin-up produces a mean circulation that is not realistic, and differs from the parent IGCM, that raises concerns about the experimental design. My concern here has to do with more clarity on how the models 2-D representation performs, particularly given the focus on specific latitudinal structures that are highlighted. These may well be correct, but the paper needs more clarity on its underlying methodology.

To recap, in this work we are considering how part of the dynamical feedback (the part represented by zonally symmetric dynamics) affects the temperature response to the radiative effects of imposed annual cycles in trace species. We do this by comparing a control configuration, in which there is no imposed annual cycle in trace species, with a perturbed configuration, in which the imposed annual cycle has been added. The reviewers questions have in essence focused on the response we deduce and how our conclusions depend on the control configuration of the model. The key issue here is not the control configuration itself, but whether differences in the control configuration (for example, the 2K difference between the control configuration described in the paper and ERA-I annual temperatures) lead to substantial differences in the response.

We had investigated this sensitivity previously – statements such as 'differences from the ERA-Interim annual mean are small (less than 2K in the tropical stratosphere) and do not affect the results presented below' included in previous versions of the paper were based on results from sensitivity experiments, and we have investigated it further at this stage by including a more realistic upwelling in the model control state. Once again our finding is that differences in the temperature response are small, e.g., a maximum of 0.2K, and that our characterisation of the latitudinal structure of the response continues to apply.

To ensure that our approach and the assumptions within are as clear as possible we have further revised the text of the paper (page 17, line 31) and we hope that this further addresses the reviewers concerns (e.g., over the need for a spin-up period in the control and perturbed simulations).

4) I appreciate the added appendix C but remain concerned about the discussion of uncertainties in SWOOSH ozone, and the authors' response to my criticism. The authors use the interannual standard deviation of SWOOSH values in providing tests in Appendix C. But the variability from year to year is not a measure of absolute accuracy. This should be explicitly stated. Further, the authors quote Tummon et al. (page 29, line 30) regarding the spread among satellites in the lower stratosphere. But it is clear that the uncertainties in the region of interest here, the lowermost stratosphere in the tropics, are considerably greater than global average uncertainties, in part because ozone is so much less abundant in the tropical lower stratosphere than globally. In the response to my comment, the authors assert an uncertainty of 10% at 70 hPa

and 30-40% by 130 hPa and refer to Davis et al. I could find no such statement for the tropics in the Davis et al. paper. Again, I request the authors to clarify where specifically this estimate comes from, whether it is a value for the tropics or simply a global average and therefore probably too generous in the tropics. Please give uncertainties appropriate to tropical latitudes and say where they came from.

Two sets of values presented in the paper are intended to give the reader a sense both the precision and the accuracy of the dataset and are appropriate to tropical latitudes. Appendices B and C have been edited to add details to clarify the calculation being done.

Precision: The 10% at 70 hPa and 30-40% by 130 hPa is not in the Davis et al paper but inferred from the SWOOSH dataset following the treatment of uncertainties as described in Davis et al. and provided by the SWOOSH dataset. The method by which these are calculated are detailed in Davis et al. (2016). This is an estimate of the precision of the dataset and is in line with similar estimates of precision at these levels in AURA MLS satellite data in the tropics (Livesey et al., 2017).

Accuracy: The tests in Appendix C are illustrative calculations and this is mentioned in the text. These values are taken as the interannual standard deviation of SWOOSH values to obtain a meaningful ozone perturbation in a way such that the calculations are reproducible. The variability is not an measure of accuracy but the range of values used reflects the spread in the accuracy of the SWOOSH dataset. This is comparable to the AURA MLS estimate of the accuracy compared to tropical sonde data given in Table 3.18.1 of Livesey et al. (2017) (Also, Sean Davis, pers. comm.).

5) page 5 and elsewhere. Temperatures are taken from ERA for 1991-2010, but ozone and water vapor from SWOOSH are for 1984-2015. Why the inconsistency? Why not use 1991-2010 from SWOOSH?

The time series is long enough to produce reasonable climatologies. E.g., the maximum differences between using a climatology from 1991 to 2010 for both ozone and water vapour from SWOOSH and the 1984 to 2015 SWOOSH climatology are of an order of magnitude smaller than the amplitude of the annual cycle perturbations in the region of interest.

6) Page 5, lines 28-31. The assertion that reproducing the annual cycle means cloud effects must be small should be removed for the reasons discussed above.

This comment has been addressed together with point 2) above.

> 7) page 6, line 12. What is meant by 'accurate convergence' and how was this determined? Loose language detracts from the paper in several places; this is an example but I suggest that the authors try to tighten up throughout.

The meaning of 'Accurate convergence' is explained in Appendix A and a reference to this appendix has been added to the main text (page 6, line 5).

> 8) page 8, line 12-16. I think 10% is far smaller than can be defended as a 2-sigma absolute (not interannual) uncertainty for tropical lower stratospheric ozone. This sentence is not useful or needed, just delete it.

We are arguing, together with the comment above to point 4), that 10% can be defended as the 2-sigma precision of the dataset for the annual cycle variation in ozone at 70hPa. We are not attempting to quantify exactly the absolute uncertainty of the dataset but an indication of this can be obtained from comparisons to the accuracy of the various satellite instruments. To address this, we have instead presented a range of calculations in Appendix C.

> 9) page 8, line 23. You quote 60% from one region and 30% from another. Where is the remaining 10%? Is there a 40% total contribution when additional higher levels are considered?

There is an 8% contribution from the region between 30 to 50 hPa region with the remainder coming from the remaining region. The text has been edited to add this (page 8, line 15).

> 10) Pages 6-10. The paper uses loose and inconsistent language in deciding when there is a 'significant' non-local contribution for water vapor versus ozone. In the case of water vapor, on page 10, line 7, an 0.4K non-local contribution for water vapor out of 1.1K (i.e., 36%) is deemed 'significant', while on page 8, line 22, a 30% contribution from ozone from higher altitudes is simply dismissed (with 10% unaccounted for), and it is stated here and elsewhere that

the ozone response is primarily local. Please do account for the missing 10%
by examining your calculations rather than leaving this hanging. In any case,
even without the extra 10%, it looks to me like the two are comparable, and this
should be stated. I suggest that the word 'significant' may be best reserved for
things tested statistically, not those where the authors view is expressed. The
paper should avoid loose judgments on what is 'significant' and what is not, and
make consistent comparisons throughout. The question of non-local effects for
both ozone and water vapor comes up in several other places in the paper, and
needs consistent propagation in all its places of occurrence.

The 30% contribution to the temperature at 90hPa from the ozone annual cycle in the region between 80 to 50 hPa is indeed significant and has not been dismissed. The remaining contribution has now been accounted for in the edited text. It is expected that there will be a substantial contribution from the region above 90hPa to the temperature change there since there is a large annual cycle of ozone in this region.

The distinction being made between ozone and water vapour relates to the regions where these trace gases cause their largest temperature response. For ozone this region is located around 70hPa and for this level, 80% of the temperature annual cycle due to ozone comes from ozone variations local to this region. In contrast, temperature due to water vapour maximise around 90hPa and only 60% of this comes from the water vapour in the 100 to 80hPa region. This has been clarified in the discussion (page 22, line 10).

When comparisons are made, we have also replaced the word 'significant' with the word 'substantial'.

11) Page 11, line 6. You dont compare your findings regarding non-local effects to Gilford and Solomon. That should be done here, since it was a key focus of that paper.

We have added a comparison to the GS2017 results and added the following to the text (page 11, line 25):

"Gilford and Solomon (2017) find a response to water vapour changes with a peak-to-peak amplitude of $0.6\,K$ at $70\,hPa$, $0.9\,K$ at $85\,hPa$ and $0.5\,K$ at $100\,hPa$. These values are smaller than the amplitudes (respectively $0.9\,K$, $1.1\,K$ – for $90\,hPa$ and $1.0\,K$) we report above, particularly at $100\,hPa$ but the difference may be in part explained by the fact that our calculations include water vapour variations down to $130\,hPa$. When, following Gilford and Solomon (2017), we include water vapour variations only above $117\,hPa$, we obtain peak-to-peak amplitudes of $0.8\,K$, $1\,K$ (for $85\,hPa$) and $0.8\,K$ respectively, closer to their results."

12) Page 14, lines 9-13. This statement first critiques Randel, then speculates as to why his results may differ, then admits to substantial uncertainties underlying that criticism. Avoid loose speculation in critiquing other work, delete this.

The statement is not aimed at critiquing Randel, rather we are trying to compare our calculation to theirs and point out physical reasons why our results differ. We identify in this section the importance of a tropospheric constraint, the effects of which were implicitly included but not explicitly identified by the Randel et al. paper. The text has been modified as follows to make this clearer (page 14, line 8):

"Randel et al. (2002) inferred damping time scales from the cross-correlation between the annual components of analysed $\overline{T}$ and $\overline{w}^*$, which implicitly includes non-local effects such as those of non-radiative processes operating in the upper troposphere. This is also true of the supporting radiative calculations they performed on the basis of observed temperature anomalies. As demonstrated below, the tropospheric processes have a substantial effect on the relaxation of temperature anomalies even in the lower stratosphere, in part because of the strong dependence of radiative timescales on the vertical scale of the imposed temperature perturbation (Fels, 1982)."

13) Page 15 and elsewhere. This section is interesting, but the paper is too vague regarding the outcome. What is the origin of the implied "upper tropospheric constrain"? Could it be clouds? Something else? Are there any insights to be had from the full IGCM? This isnt very useful unless it can at least clarify what may be meant here.

The main message of this section is to show the various factors that determine the vertical structure of the temperature annual cycle. The 'upper tropospheric constraint' is justified as being necessary from the small observed annual cycle below 130hPa in ERA-Interim. We did not explore what causes this constraint (and we said that) but we have presented clear arguments as to how the existence of the constraint has been deduced. We see this material as important precisely because it raises interesting questions that cannot be answered and we have retained it.

For clarification we have added the following to the text on Page 23, line 19:

"We have not attempted to provide an explanation for the inferred upper-tropospheric constraint and highlight this as an area for further study."

14) page 22, lines 23-26. Ozone values are stated for 70 hPa, while H2O values are stated for 90 hPa. Consistent comparisons should be made. Page 8, lines 21-24 state that 30% of the ozone effect at 90 mbar is non-local, (perhaps 40%, when the missing 10% is accounted for). Therefore, at 90 hPa your results suggest similar magnitudes for non-local effects of both species.

The text has been edited and this comparison has been clarified together with point 10) above.

15) Please comment on the differences between Fig 1 (ERA-interim) and the combined effects of ozone and water in the IGCM shown in Fig 12. Do the residuals seem plausible? What do you think accounts for them? At 90 hPa in particular, there seem to be some large remaining terms if this model is correct. What do you think they could be?

As explained at various places in the paper – e.g., on Page 17, Line 3, the dynamical model (IGCM) calculation shown in Fig 12 is not intended to reproduce the full ERA-Interim temperature change, since the model calculates only the zonally symmetric dynamical adjustment to the ozone and water vapour radiative heating without a full treatment of the role of wave induced forces driving the annual cycle. (The implications of this are discussed in the final Discussion section.)

**Bibliography**

Davis, S. M., and Coauthors, 2016: The Stratospheric Water and Ozone Satellite Homogenized (SWOOSH) database: A long-term database for climate studies. *Earth System Science Data Discussions*, 1–59, doi:10.5194/essd-2016-16, URL http://www.earth-syst-sci-data-discuss.net/essd-2016-16/.

Fels, S. B., 1982: A Parameterization of Scale-Dependent Radiative Damping Rates in the Middle Atmosphere. *Journal of the Atmospheric Sciences*, **39**, 1141–1152.

Gilford, D. M., and S. Solomon, 2017: Radiative effects of stratospheric seasonal cycles in the tropical upper troposphere and lower stratosphere. *Journal of Climate*, JCLI–D–16–0633.1, doi:10.1175/JCLI-D-16-0633.1, URL http://journals.ametsoc.org/doi/10.1175/JCLI-D-16-0633.1.

Livesey, N. J., and Coauthors, 2017: Earth Observing System (EOS) Version 4.2x-3.0 Level data quality and description document. Tech. rep.

Randel, W. J., R. R. Garcia, and F. Wu, 2002: Time-Dependent Upwelling in the Tropical Lower Stratosphere Estimated from the Zonal-Mean Momentum Budget. *Journal of the Atmospheric Sciences*, **59 (13)**, 2141–2152, doi:10.1175/1520-0469(2002)059⟨2141:TDUITT⟩2.0.CO;2, URL http://journals.ametsoc.org/doi/abs/10.1175/1520-0469(2002)059{\%}3C2141:TDUITT{\%}3E2.0.CO;2.